# FSW-GNN: A Bi-Lipschitz WL-Equivalent Graph Neural Network

## ABSTRACT

Many of the most popular graph neural networks fall into the category of message-passing neural networks (MPNNs). Famously, MPNNs' ability to distinguish between graphs is limited to graphs separable by the Weisfeiler-Lemann (WL) graph isomorphism test, and the strongest MPNNs, in terms of separation power, are WL-equivalent.

Recently, it was shown that the quality of separation provided by standard WL-equivalent MPNN can be very low, resulting in WL-separable graphs being mapped to very similar, hardly distinguishable features. This paper addresses this issue by seeking bi-Lipschitz continuity guarantees for MPNNs. We demonstrate that, in contrast with standard summation-based MPNNs, which lack bi-Lipschitz properties, our proposed model provides a bi-Lipschitz graph embedding with respect to two standard graph metrics. Empirically, we show that our MPNN is competitive with standard MPNNs for several graph learning tasks and is far more accurate in over-squashing long-range tasks.

## 1 INTRODUCTION

Graph neural networks are a central research topic in contemporary machine learning research. Many of the most popular models, such as GIN (Xu et al., 2019), GraphSage (Hamilton et al., 2017), GAT (Velickovic et al., 2018), and GCN (Kipf & Welling, 2017), can be seen as an instantiation of Message Passing Neural Networks (MPNNs) (Gilmer et al., 2017).

A well-known limitation of MPNNs is that they cannot differentiate between all distinct pairs of graphs. In fact, a pair of distinct graphs that cannot be separated by the Weisfeiler-Lehman (WL) graph isomorphism test will not be separated by *any* MPNN (Xu et al., 2019). Accordingly, the most expressive MPNNs are those that are *WL-equivalent*, which means they can separate all pairs of graphs that are separable by WL. WL-equivalent MPNNs were proposed in the seminal works of Xu et al. (2019); Morris et al. (2019), and the complexity of these constructions was later improved in (Aamand et al., 2022; Amir et al., 2023).

While separation should theoretically be achieved for WL-equivalent MPNNs, in some cases, their separation is so weak that it cannot be observed with 32-bit floating-point computer numbers (see Bravo et al. (2024b)). This observation motivates the development of *quantitative* estimates of MPNN separation by means of bi-Lipschitz stability guarantees. These guarantees would ensure that Euclidean distances in the MPNN feature space are neither much larger nor much smaller than distances in the original graph space, which are defined by a suitable graph metric (defined up to WL equivalence).

Some first steps towards addressing these challenges have already been made by Davidson & Dym (2024). Their work analyzes a weaker notion of Lipschitz and Holder guarantees *in expectation*, and shows that essentially all popular MPNN models are not lower-Lipschitz, but they are lower-Holder in expectation, with an exponent that grows worse as the MPNN depth increases. In contrast, they propose SortMPNN, a novel MPNN which *is* bi-Lipschitz (in expectation).

However, SortMPNN has several limitations. First, it is only bi-Lipschitz in expectation–a relaxed notion of Lipschitzness that guarantees smoothness only in expectation over the model parameters and for fixed pairs of graphs rather than uniformly on all input graphs. Additionally, their method addresses neighborhoods of different sizes by augmenting them to a predetermined maximum size.

This approach has significant limitations: it is both computationally expensive, as the model cannot exploit graph sparsity, and it necessitates prior knowledge of the maximal graph size for the learning task at hand.

In this paper we introduce a novel MPNN called FSW-GNN (Fourier Sliced-Wasserstein GNN), which overcomes the limitations of SortMPNN. We show that this model is Bi-Lipschitz in the standard sense rather than in expectation, with respect to both the DS metric of Grohe (2020) and the Tree Mover's Distance (TMD) metric of Chuang & Jegelka (2022). Furthermore, this model can handle sparsity well and thus is much more efficient than SortMPNN for sparse graphs.

Empirically, we show that FSW-GNN has comparable or better performance than MPNN on 'standard' graph learning tasks, but achieves superior performance when considering long-range tasks, which require a large number of message-passing iterations. We hypothesize this is because the Holder exponent of standard MPNNs deteriorates with depth, whereas our FSW-GNN is bi-Lipschitz for any finite number of iterations. This hypothesis provides an alternative, and perhaps complementary, explanation to the difficulty of training deep standard MPNNs, commonly attributed to over smoothing and over squashing .

### 1.1 RELATED WORKS

**MPNNs with advanced pooling mechanisms**  In addition to the SortMPNN model discussed earlier, our approach is conceptually related to other MPNNs that replace basic max-, mean-, or sum-pooling with more advanced pooling methods, such as sorting (Balan et al., 2022; Zhang et al., 2018), standard deviation (Corso et al., 2020), or Wasserstein embeddings via reference distributions (Kolouri et al.). However, these methods lack the bi-Lipschitzness guarantees that our model provides.

**Bi-Lipschitzness**  Bi-Lipschitzness guarantees arise naturally in many domains, including frames (Balan, 1997), phase retrieval (Bandeira et al., 2014; Cheng et al., 2021), group invariant learning (Cahill et al., 2020; 2024b) and multisets (Amir et al., 2023; Amir & Dym, 2024). In the context of MPNNs, a recent survey by Morris et al. (2024) identifies bi-Lipschitzness guarantees as a significant future challenge for theoretical GNN research. While most MPNNs are upper Lipschitz, as discussed in (Chuang & Jegelka, 2022; Levie, 2023; Davidson & Dym, 2024), achieving bi-Lipschitzness remains an open problem.

**WL-equivalent metrics**  Metrics with the separation power of WL include the DS metric (Grohe, 2020) (also called the tree metric), the TMD metric (Chuang & Jegelka, 2022), and the WL metric (Chen et al., 2022). In this paper, we prove that the graph embeddings computed by our FSW-GNN model are bi-Lipschitz with respect to the DS and TMD metrics. This analysis is for graphs with bounded cardinality and continuous, bounded features.  Weaker notions of equivalence between these metrics, in the setting of graphs with unbounded cardinality and without node features, are discussed in (Böker et al., 2024; Böker, 2021).

## 2 PROBLEM SETTING

In this section, we outline the problem setting, first providing the theoretical background of the problem and then stating our objectives.

**Vertex-featured graphs**  Our main objects of study are graphs with vertex features, represented as triplets $G = (V, E, X)$, where $V = \{v_i\}_{i=1}^n$ is the set of vertices, $E \subseteq \{\{v_i, v_j\} \mid i, j \in [n]\}$ is the set of undirected edges in $G$, and $X = [\boldsymbol{x}_1, \ldots, \boldsymbol{x}_n]$ is a matrix containing the vertex feature vectors $\boldsymbol{x}_i \in \Omega$, where the *feature domain* $\Omega$ is a subset of $\mathbb{R}^d$. We denote by $\mathcal{G}_{\leq N}(\Omega)$ the set of all vertex-featured graphs with at most $N$ vertices and corresponding features in $\Omega$. Throughout the paper, we use $\{\}$ to denote multisets.

**Weisfeiler-Lemann Graph Isomorphism test**  Two graphs are *isomorphic* if they are identical up to relabeling of their nodes. Perhaps surprisingly, the problem of determining whether two given graphs are isomorphic is rather challenging. To date, no known algorithm can solve it in polynomial

time (Babai, 2016). However, there exist various heuristics that provide an incomplete but often adequate method to test whether a given pair of graphs is isomorphic. The most notable example is the Weisfeiler-Leman (WL) graph isomorphism test.

The WL test can be described as assigning to each graph $G = (V, E, X)$ a feature vector $c_G^T$ according to the following recursive formula.

$$
\begin{aligned}
c_v^0 &:= \mathbf{X}_v, \quad v \in V, \\
c_v^t &:= \text{Combine}\big(c_v^{t-1}, \text{Aggregate}\big(\big\{c_u^{t-1} \mid u \in \mathcal{N}_v\big\}\big)\big), \quad 1 \le t \le T, \\
c_G^T &:= \text{Readout}\big(\big\{c_{v_1}^T, \ldots, c_{v_n}^T\big\}\big),
\end{aligned}
\tag{1}
$$

where Aggregate and Readout are functions that injectively map multisets of vectors in Euclidean space into another Euclidean space, Combine is an injective function from one Euclidean space to another, and $\mathcal{N}_v$ denotes the neighborhood of the vertex $v$ in $G$.

**Definition** (WL graph equivalence)**.** Two vertex-featured graphs $G$ and $\tilde{G}$ are said to be *WL-equivalent*, denoted by $G \overset{\text{WL}}{\sim} \tilde{G}$, if $c_G^T = c_{\tilde{G}}^T$ for all $T \ge 0$. Otherwise, they are said to be *WL-separable* or *WL-distinguishable*.

It is a known fact (Grohe, 2021; Morris et al., 2023) that for $G, \tilde{G} \in \mathcal{G}_{\le N}(\mathbb{R}^d)$, if the equality $c_G^T = c_{\tilde{G}}^T$ is satisfied for $T = N$, then it is satisfied for all $T \ge 0$, and thus $G \overset{\text{WL}}{\sim} \tilde{G}$.

While the WL test can distinguish most pairs of non-isomorphic graphs, there exist examples of non-isomorphic graph pairs that WL cannot separate; see (Zopf, 2022). Note that there exist higher-order versions of this test called $k$-WL, $k \ge 2$, but in this paper, we consider only the 1-WL test, denoted by WL for brevity.

**Message passing neural networks**  Message Passing Neural Networks (MPNNs) are the most common neural architectures designed to compute graph functions. They operate on a similar principle to the WL test, but with the purpose of performing predictions on graphs rather than determining if they are isomorphic. Their core mechanism is the message-passing procedure, which maintains a hidden feature for each vertex and iteratively updates it as a function of the neighbors' features. This process is outlined as follows:

1. **Initialization:**  The hidden feature $\boldsymbol{h}_v^0$ of each node is initialized by its input feature $\boldsymbol{x}_v$.

2. **Message aggregation:** Each node $v \in V$ aggregates messages from its neighbors by

$$
m_v^{(t)} := \text{Aggregate}\Big(\Big\{\boldsymbol{h}_u^{(t-1)} \mid u \in \mathcal{N}_v\Big\}\Big)
\tag{2}
$$

   Where Aggregate is a multiset-to-vector function.

3. **Update step:** Each node updates its own hidden feature according to its aggregated messages and its previous hidden feature, using a vector-to-vector *update function*:

$$
\boldsymbol{h}_v^{(t)} := \text{Update}\Big(m_v^{(t)}, \boldsymbol{h}_v^{(t-1)}\Big),
\tag{3}
$$

4. **Readout:** After $T$ iterations of steps 2-3, a global graph-level feature $\boldsymbol{h}_G$ is computed from the multiset of hidden features $\Big\{h_v^{(T)} \mid v \in V\Big\}$ by a *readout function*:

$$
\boldsymbol{h}_G := \text{Readout}\Big(\Big\{\boldsymbol{h}_v^{(T)} \mid v \in V\Big\}\Big).
$$

Numerous MPNNs were proposed in recent years, including GIN (Xu et al., 2019), GraphSage (Hamilton et al., 2017), GAT (Velickovic et al., 2018), and GCN (Kipf & Welling, 2017), the main differences between them being the specific choices of the aggregation, update, and readout functions. An MPNN computes an *embedding* $F(G) = \boldsymbol{h}_G$, which maps the graphs in $\mathcal{G}_{\le N}(\Omega)$ to vectors in $\mathbb{R}^m$. The obtained embedding is often further processed by standard machine-learning tools for vectors, such as multi-layer perceptrons (MLPs), to obtain a final graph prediction. The ability of such a model to approximate functions on graphs is closely related to the separation properties of $F$. If $F$ can differentiate between any pair of non-isomorphic graphs, then a model of the form $\text{MLP} \circ F$ would be able to approximate any functions on graphs Chen et al. (2019).

Unfortunately, MPNN cannot separate any pair of WL-equivalent graphs, even if they are not truly isomorphic Xu et al. (2019); Morris et al. (2019). Accordingly, the best we can hope for from an MPNN, in terms of separation, is *WL equivalence*: for every pair of graphs $G, G' \in \mathcal{G}_{\leq N}(\Omega)$, $F(G) = F(G')$ *if and only if* $G \overset{\text{WL}}{\sim} G'$. While MPNNs based on max- or mean-pooling cannot be WL-equivalent Xu et al. (2019), it is possible to construct WL-equivalent MPNNs based on sum-pooling, as discussed in Xu et al. (2019); Morris et al. (2019); Aamand et al. (2022); Amir et al. (2023); Bravo et al. (2024a). Theoretically, a properly tuned graph model based on a WL-equivalent MPNN should be capable of perfectly solving any binary classification task, provided that no two WL-equivalent graphs have different ground-truth labels. However, this separation does not always manifest in practice. One reason is that WL-equivalent functions may map two input graphs far apart in the input space to outputs that are numerically indistinguishable in the output Euclidean space. In fact, Davidson & Dym (2024) provides an example of graph pairs that are not WL-equivalent yet are mapped to near-identical outputs by standard sum-based MPNNs. Consequently, these MPNNs fail on binary classification tasks for such graphs.

This paper aims to address this limitation by devising an MPNN whose embeddings preserve distances in the original graph space in the bi-Lipschitz sense. To state our goal formally, we need to define appropriate notions of WL metrics on the input space and bi-Lipschitz graph embeddings.

**WL-metric for graphs** WL-metrics quantify the *extent* to which two graphs are not WL-equivalent:

**Definition** (WL-metric). A *WL-metric on* $\mathcal{G}_{\leq N}(\Omega)$ is a function $\rho : \mathcal{G}_{\leq N}(\Omega) \times \mathcal{G}_{\leq N}(\Omega) \to \mathbb{R}_{\geq 0}$ that satisfies the following conditions for all $G_1, G_2, G_3 \in \mathcal{G}_{\leq N}(\Omega)$:

$$\rho(G_1, G_2) = \rho(G_2, G_1) \qquad \textit{Symmetry} \qquad (4a)$$

$$\rho(G_1, G_3) \leq \rho(G_1, G_2) + \rho(G_2, G_3) \qquad \textit{Triangle inequality} \qquad (4b)$$

$$\rho(G_1, G_2) = 0 \iff G_1 \overset{\text{WL}}{\sim} G_2. \qquad \textit{WL equivalence} \qquad (4c)$$

Note that strictly speaking, such $\rho$ is a *pseudo-metric* on $\mathcal{G}_{\leq N}(\mathbb{R}^d)$ rather than a metric, as it allows distinct graphs to have a distance of zero if they are WL-equivalent. However, we will use the term *metric* to denote a pseu-dometric for convenience.

In this paper, we consider two WL metrics: The first is the DS metric, proposed in (Grohe, 2021). Originally, this metric was defined only for featureless graphs of the same cardinality. In the next section, we will discuss extending it to the more general case of $\mathcal{G}_{\leq N}(\mathbb{R}^d)$. The second WL metric we consider is the Tree Mover's distance (TMD). This metric was proposed and shown to be a WL-metric by Chuang & Jegelka (2022).

**Bi-Lipschitzness** Once a WL-metric is defined to measure distances between graphs, one can bound the distortion incurred by a graph embedding with respect to that metric, using the notion of bi-Lipschitzness:

**Definition** (Bi-Lipschitz embedding). Let $\rho$ be a WL-metric on $\mathcal{G}_{\leq N}(\Omega)$. An embedding $E : \mathcal{G}_{\leq N}(\Omega) \to \mathbb{R}^m$ is said to be *bi-Lipschitz with respect to* $\rho$ *on* $\mathcal{G}_{\leq N}(\Omega)$ if there exist constants $0 < c \leq C < \infty$ such that

$$c \cdot \rho(G_1, G_2) \leq \|E(G_1) - E(G_2)\|_2 \leq C \cdot \rho(G_1, G_2), \quad \forall G_1, G_2 \in \mathcal{G}_{\leq N}(\Omega). \qquad (5)$$

If $E$ satisfies just the left- or right-hand side of (5), it is said to be *lower-Lipschitz* or *upper-Lipschitz*, respectively.

Bi-Lipschitzness ensures that the embedding maps the original space $\mathcal{G}_{\leq N}(\Omega)$ into the output Euclidean space with bounded distortion, with the ratio $\frac{C}{c}$ acting as an upper bound on the distortion, akin to the condition number of a matrix. This enables the application of metric-based learning methods, such as clustering and nearest-neighbor search, to non-Euclidean input data. This is discussed, for example, in (Cahill et al., 2024a).

**Lipschitzness, Holder, and depth** In the context of graph neural networks, we conjecture that the advantage of bi-Lipschitz MPNNs over standard sum-based MPNN will be more apparent for

'deep' MPNNs, where the number $T$ of message-passing iterations is large. Our reasoning for this conjecture can be explained via the related notion of lower-Holder MPNN.

**Definition.** A graph embedding is *lower-Holder* with constants $c > 0$, $\alpha \geq 1$, if

$$c \cdot \rho(G_1, G_2)^\alpha \leq \|E(G_1) - E(G_2)\|_2 \quad \forall G_1, G_2 \in \mathcal{G}_{\leq N}(\Omega). \tag{6}$$

In general, the larger $\alpha$ is, the worse the distortion. The best case of $\alpha = 1$ coincides with lower-Lipschitzness. Davidson & Dym (2024) showed that standard sum-based MPNNs are lower-Holder (in expectation), with an exponent $\alpha$ the becomes worse as the number $T$ of message-passing iterations increases. This means that the worst-case distance between sum-based graph embeddings can go to zero super-exponentially with $T$. In contrast, our bi-Lipschitz MPNN will remain bi-Lipschitz for any finite $T$ (although the distortion $C/c$ may depend on $T$), and hence will be more robust to increasing the number of message-passing iterations.

The challenge of training deep MPNNs is one of the core problems in graph neural networks (Morris et al., 2024). The difficulty in doing so is often attributed to oversmoothing (Rusch et al., 2023) or oversquashing (Alon & Yahav, 2020). Based on our results, we conjecture that distortion of the graph metric may be the root of this problem, and that, as a result, bi-Lipschitz MPNNs are a promising solution. We provide empirical evidence for this conjecture in Section 4, where we show that our bi-Lipschitz MPNN is far superior to standard MPNNs on long range tasks which require training a deep MPNN.

## 3 MAIN CONTRIBUTIONS

In this section, we discuss our main contributions. We begin by defining our generalized DS metric. We will then discuss our MPNN and FSW-GNN and show that it is bi-Lipschitz with respect to both DS and TMD.

**The DS metric**  The roots of the DS metric come from a relaxation of the graph isomorphism problem. Two graphs $G$ and $\tilde{G}$, each with $n$ vertices, and corresponding adjacency matrices $A$ $\tilde{A}$ are isomorphic if and only if there exists a *permutation matrix* $P$ such that $AP = P\tilde{A}$. Since checking whether graphs are isomorphic is intractable, an approximate solution can be sought by considering the equation $AS = S\tilde{A}$, where $S$ is a matrix in the convex hull of the permutation matrices: the set of doubly stochastic matrices, denoted by $\mathcal{D}_n$. These are $n \times n$ matrices with non-negative entries whose rows and columns all sum to one. Remarkably, this equation admits a doubly stochastic solution if and only if the graphs are WL-equivalent (Scheinerman & Ullman, 2013). Accordingly, a WL-metric can be defined by the minimization problem.

$$\rho_{\text{DS}}\left(G, \tilde{G}\right) = \min_{S \in \mathcal{D}_n} \left\|AS - S\tilde{A}\right\|_2, \tag{7}$$

where $\|\cdot\|_2$ denoted the entry-wise $\ell_2$ norm for matrices. The optimization problem in (7) can be solved by off-the-shelf convex optimization solvers and was considered as a method for finding the correspondence between two graphs in many papers, including Aflalo et al. (2015); Lyzinski et al. (2016); Dym (2018); Dym et al. (2017); Bernard et al. (2018).

The idea of using the DS metric for MPNN stability analysis was introduced in (Grohe, 2020) and further discussed by Böker (2021). To apply this idea to our setting, we need to adapt this metric to vertex-featured graphs with varying numbers of vertices. We do this by augmenting it as follows:

$$\rho_{\text{DS}}\left(G, \tilde{G}\right) = |n - \tilde{n}| + \min_{S \in \Pi(n, \tilde{n})} \left\|AS - S\tilde{A}\right\|_2 + \sum_{i \in [n], j \in [\tilde{n}]} S_{ij}\|\boldsymbol{x}_i - \tilde{\boldsymbol{x}}_j\|_2, \tag{8}$$

where $n$ and $\tilde{n}$ denote the number of vertices in $G$ and $\tilde{G}$, $\boldsymbol{x}_i$ and $\tilde{\boldsymbol{x}}_j$ denote the vertex features of $G$ and $\tilde{G}$, and $\Pi(n, \tilde{n})$ is the set of $n \times \tilde{n}$ matrices $S$ with non-negative entries, that satisfy $\sum_{j=1}^{\tilde{n}} S_{ij} = \frac{1}{n}$, $\sum_{i=1}^{n} S_{ij} = \frac{1}{\tilde{n}}$

**Theorem 3.1.** [Proof in Appendix C.1] *Let* $\rho_{\text{DS}} : \mathcal{G}_{\leq N}\left(\mathbb{R}^d\right) \times \mathcal{G}_{\leq N}\left(\mathbb{R}^d\right) \to \mathbb{R}_{\geq 0}$ *be as in* (8). *Then* $\rho_{\text{DS}}$ *is a WL-equivalent metric on* $\mathcal{G}_{\leq N}\left(\mathbb{R}^d\right)$.

**Bi-Lipschitz MPNN** We now present our main contribution: a novel MPNN that is not only WL-equivalent but also bi-Lipschitz, both with respect to the metric $\rho_{\text{DS}}$ and TMD.

The core innovation in our MPNN lies in its message aggregation method. To aggregate messages, we use the *Fourier Sliced-Wasserstein (FSW) embedding*-a method for embedding multisets into Euclidean space, proposed by Amir & Dym (2024), where it was shown to be bi-Lipschitz. Consequently, it seems plausible a priori that an MPNN based on FSW aggregations will be bi-Lipschitz for *graphs*. In the following, we prove that this is indeed the case. We begin by describing the FSW embedding and then introduce our FSW-GNN architecture.

The FSW embedding maps input multisets $\{\boldsymbol{x}_1, \ldots, \boldsymbol{x}_n\}$, with $\boldsymbol{x}_1, \ldots, \boldsymbol{x}_n \in \mathbb{R}^d$, to output vectors $\boldsymbol{z} = (z_1, \ldots, z_m) \in \mathbb{R}^m$. In addition to the input, it depends on parameters $\boldsymbol{v}_i \in \mathcal{S}^{d-1}$ and $\xi_i \in \mathbb{R}$, representing projection vectors and frequencies; see (Amir & Dym, 2024) for details. It is denoted by

$$E_{\text{FSW}}(\{\boldsymbol{x}_1, \ldots, \boldsymbol{x}_n\}; \ (\boldsymbol{v}_i, \xi_i)_{i=1}^m) = \boldsymbol{z}.$$

The $i$-th coordinate of the output $\boldsymbol{z}$ is a scalar $z_i$, defined by the formula

$$\boldsymbol{y}_i = \text{sort}\left(\boldsymbol{v}_i \cdot \boldsymbol{x}_1, \ldots, \boldsymbol{v}_i \cdot \boldsymbol{x}_n\right) \tag{9}$$

$$Q_{\boldsymbol{y}_i}(t) = \sum_{j=1}^n y_{i,j} \chi_{\left[\frac{i-1}{n}, \frac{j}{n}\right)}(t) \tag{10}$$

$$z_i = 2(1 + \xi) \int_0^1 Q_{\boldsymbol{y}_i}(t) \cos(2\pi\xi t) dt \tag{11}$$

where in this formula $\boldsymbol{x} \cdot \boldsymbol{y}$ denotes the standard inner product of $\boldsymbol{x}$ and $\boldsymbol{y}$, the function $\chi_{[a,b)}$ is the indicator function of the interval $[a, b)$, and $y_{i,j}$ is the $j$-th entry of the vector $\boldsymbol{y}_i$.

The FSW embedding is essentially computed in three steps: first, a direction vector $\boldsymbol{v}_i$ is used to project each $d$ dimensional vector to a scalar. We thus obtain a multiset of scalars which can then be sorted. This first step is the sort-type embedding used in SortMPNN, and it can be shown to be bi-Lipschitz on multisets of fixed cardinality Balan et al. (2022). However, the disadvantage of taking $\boldsymbol{y}_i$ as the embedding is that it has the same cardinality as the multiset, and so this embedding is not readily applicable to multisets of different cardinalities. The next two steps of the FSW embedding can be seen as an attempt to fix this disadvantage.

In the second step, the vector $\boldsymbol{y}_i$ is identified with a step function $Q_{\boldsymbol{y}_i}$ (the quantile function, see interpretation in (Amir & Dym, 2024)). Then, in the third step, the *cosine transform*, a variant of the Fourier transform, is applied to $Q_{\boldsymbol{y}_i}$, at the given frequency $\xi_i$, to obtain the final value $z_i$. Note that the integral in equation 11 has a closed form solution, and the whole procedure can be computed with complexity linear in $n, d$. Moreover, the dimension of the parameters and output of the embedding does not depend on $n$, and thus this embedding is suitable for multisets of varying sizes.

**FSW-GNN** The FSW-GNN model processes input graphs $G = (V, E, X)$ by $T$ message-passing iterations according to the following recursive formula:

$$\begin{aligned}
\boldsymbol{h}_v^{(0)} &:= \boldsymbol{x}_v, \\
\boldsymbol{q}_v^{(t)} &:= E_{\text{FSW}}^{(t)}\left(\left\{\boldsymbol{h}_u^{(t-1)} \mid u \in \mathcal{N}_v\right\}\right), \quad 1 \le t \le T \\
\boldsymbol{h}_v^{(t)} &:= \Phi^{(t)}\left(\left[\boldsymbol{h}_v^{(t-1)}; \boldsymbol{q}_v^{(t)}\right]\right),
\end{aligned} \tag{12}$$

where the functions $E_{\text{FSW}}^{(t)}$ are all instances of the FSW embedding, $\Phi^{(t)}$ are MLPs, and $[\boldsymbol{x}; \boldsymbol{y}]$ denotes column-wise concatenation of column vectors $\boldsymbol{x}$ and $\boldsymbol{y}$. Finally, a graph-level output is computed by:

$$\boldsymbol{h}_G := \Psi \circ E_{\text{FSW}}^{\text{Glob}}\left(\left\{\boldsymbol{h}_v^{(T)} \mid v \in V\right\}\right), \tag{13}$$

where, again, $E_{\text{FSW}}^{\text{Glob}}$ is an FSW embedding, and $\Psi$ is an MLP.

The following theorem shows that with the appropriate choice of MLP sizes and number of iterations $T$, our proposed architecture is WL equivalent:

**Theorem 3.2** (Informal). [Proof in Appendix C.2] *Consider the FSW-GNN architecture for input graphs in $\mathcal{G}_{\leq N}(\mathbb{R}^d)$, with $T = N$ iterations, where $\Phi^{(t)}, \Psi$ are just linear funtions, and all features (except for input features) are of dimension $m \geq 2Nd + 2$. Then for Lebesgue almost every choice of model parameters, the graph embedding defined by the architecture is WL equivalent.*

The proof of Theorem 3.2 is based on the study of $\sigma$-subanalytic functions and the *Finite Witness Theorem*, introduced in (Amir et al., 2023).

It is worth noting that the output dimension $m$ required in practice is typically considerably lower than the one required in Theorem 3.2. This can be explained by the following fact: If all input graphs come from a ($\sigma$ sub-analytic) subset of $\mathcal{G}_{\leq N}(\mathbb{R}^d)$ with intrinsic dimension $D$ significantly lower than the ambient dimension $n \cdot d$, then it can be shown that $m = 2D + 2$ suffices for WL-equivalence.

**From separation to bi-Lipschitzness**  In general, WL-equivalence does not imply bi-Lipschitzness. As mentioned above, sum-based MPNN can be injective but are never bi-Lipschitz. In contrast, we will prove that for FSW-GNN, WL-equivalence does imply bi-Lipschitz-ness, under the additional assumption that the feature domain $\Omega$ is compact:

**Theorem 3.3.** [Proof in Appendix C.3] *Let $\Omega \subset \mathbb{R}^d$ be compact. Under the assumptions of Theorem 3.2, the FSW-GNN is bi-Lipschitz with respect to $\rho_{\mathrm{DS}}$ on $\mathcal{G}_{\leq N}(\Omega)$. If, additionally, $\Omega$ is a compact polygon that does not contain $\mathbf{0}$, then the FSW-GNN is bi-Lipschitz with respect to TMD on $\mathcal{G}_{\leq N}(\Omega)$.*

We now give a high-level explanation of the proof idea. The full proof is in the appendix. To prove Theorem 3.3, we rely on the following facts: (1) the output of FSW-GNN for an input graph $G = (V, E, X)$ is piecewise-linear with respect to the vertex-feature matrix $X$. This follows from properties of the FSW embedding functions used in (12) and (13). (2) both metrics $\rho_{\mathrm{DS}}$ and TMD can be transformed, with bounded distortion, into piecewise-linear metrics by choosing all the vector norms they employ to be the $\ell_1$ norm. The claim them follows from these observations and the following lemma:

**Lemma 3.4.** [Proof in Appendix C.3] *Let $f, g : M \to \mathbb{R}_{\geq 0}$ be nonnegative piecewise-linear functions defined on a compact polygon $M \subset \mathbb{R}^d$. Suppose that for all $x \in M$, $f(x) = 0$ if and only if $g(x) = 0$. Then there exist real constants $c, C > 0$ such that*

$$c \cdot g(x) \leq f(x) \leq C \cdot g(x), \quad \forall x \in M. \tag{14}$$

## 4  NUMERICAL EXPERIMENTS

We compare the performance of FSW-GNN with standard MPNNs and Sort-MPNN on both real-world benchmarks and synthetic long-range tasks. In addition, we evaluate the the bi-Lipschitz distortion incurred by each method. While our main baseline is MPNN models, we add for each task the state-of-the-art result to the best of our knowledge, typically achieved by more complex GNNs with higher expressive power and computational complexity.

**Empirical distortion evaluation**  First, to assess the distortion induced by each method, we conducted the following experiment, where we compared the distances induced by each embedding vs. the TMD and DS metric on a particularly challenging set of graph pairs; see Appendix D for details. Empirical estimates $\hat{C}, \hat{c}$ of the constants $C, c$ of (5) were computed, and the distortion estimate was taken as the ratio $\hat{C}/\hat{c}$. The results appear in Figure 1. As seen in the figure, our method yields considerably lower distortion than all competitors, which aligns with our theoretical guarantees.

**Trunsductive Learning**  Next, we compare FSW-GNN with GCN, GAT and Sort-MPNN for nine transductive learning tasks taken from Pei et al. (2020). As shown in Table 1, FSW-GNN outperforms the competition in six our of nine tasks. SortMPNN is a clear winner in two of the other graphs, which both have relatively large average degree (see Table 4). In addition, we include the state-of-the-art results known to us for each dataset. These results are typically achieved by strictly more powerful and computationally expensive models than any MPNN. On Cora we add Izadi et al. (2020b), for Cite Izadi et al. (2020a), for Pubm Izadi et al. (2020c), for Cham Rossi et al. (2024),

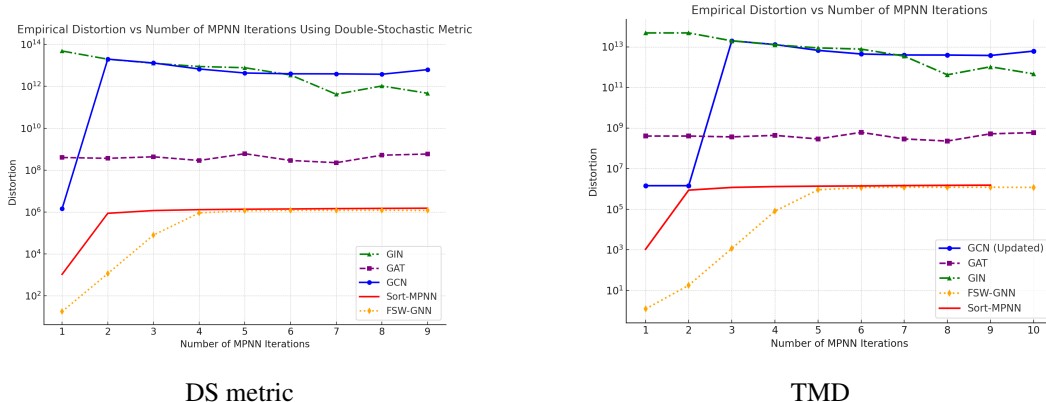

| DS metric | TMD |

Figure 1: Empirical distortion evaluation with respect to the Doubly-Stochastic (DS) Metric and Tree Mover's Distance (TMD)

for Squi Koke & Cremers (2023), for Actor Huang et al. (2024a), for Corn Eliasof et al. (2024), for Texas Luan et al. (2022), for Wisc Huang et al. (2024b) all as best known models.[1]

| Model | Cora | Cite. | Pubm. | Cham. | Squi. | Actor | Corn. | Texa. | Wisc. |
|---|---|---|---|---|---|---|---|---|---|
| GCN | 85.77 | 73.68 | 88.13 | 28.18 | 23.96 | 26.86 | 52.70 | 52.16 | 45.88 |
| GAT | **86.37** | 74.32 | 87.62 | 42.93 | 30.03 | 28.45 | 54.32 | 58.38 | 49.41 |
| FSW-GNN | 86.35 | **75.35** | **88.17** | 51.14 | 36.34 | **34.30** | 72.16 | 75.13 | 80.98 |
| Sort-MPNN | 83.46 | 72.69 | 85.15 | **78.11** | **74.69** | 31.32 | 67.03 | 70.54 | 73.92 |
| SOTA | **90.16** | **82.07** | **91.31** | **79.71** | **76.71** | **51.81** | **92.72** | **88.38** | **94.99** |

Table 1: Performance comparison across different datasets and models.

**Graph classification and regression** Here we include results on the peptides-func and peptides-struct datasets of the LRGB benchmark Dwivedi et al. (2022), which consist of graph classification and regression respectively. The results appear in Table 3. In addition, we include results on the MUTAG Debnath et al. (1991) dataset ( Table 2), where we marginally surpass other models.

| Model | MUTAG |
|---|---|
| GIN (Xu et al., 2019) | $89.4 \pm 5.6$ |
| GCN (Kipf & Welling, 2017) | $85.6 \pm 5.8$ |
| GraphSage (Hamilton et al., 2017) | $85.1 \pm 7.6$ |
| SortMPNN | $90.99 \pm 6.2$ |
| FSW-GNN | $\mathbf{91.11 \pm 7.11}$ |

Table 2: Performance of different models on the MUTAG dataset.

**Long range tasks** Next, we consider several synthetic long-range tasks suggested in the over-squashing literature Alon & Yahav (2020); Di Giovanni et al. (2023), which by design can only be solved by deep MPNN.

We first consider the NeighborsMatch problem from Alon & Yahav (2020). This node prediction problem can be solved by MPNN with $r$ iterations but not with fewer iterations. Here, $r$ is a parameter of the problem called the problem radius. The problem becomes harder as $r$ is increased. In Figure 2, we compare the performance of FSW-GNN with standard MPNN on the NeigborsMatch problem with $r$ varying from 2 to 8. Our FSW-GNN achieves perfect accuracy for all values of $r$, while Sort-MPNN fails at $r = 8$ and the other competing methods falter for $r \geq 6$. Next, we

---
[1]Our code will be made available to the public upon paper acceptance.

| Dataset | peptides-func (AP↑) | peptides-struct (MAE↓) |
|---|---|---|
| GINE Hu* et al. (2020) | 0.6621±0.0067 | 0.2473±0.0017 |
| GCN Kipf & Welling (2017) | 0.6860±0.0050 | **0.2460±0.0007** |
| GatedGCN Bresson & Laurent (2018) | 0.6765±0.0047 | 0.2477±0.0009 |
| SortMPNN Davidson & Dym (2024) | **0.6914±0.0056** | 0.2494±0.0021 |
| FSW-GNN | 0.6864±0.0048 | 0.2489±0.00155 |
| Best model | **0.73** | **0.242** |

Table 3: LRGB results. Best in **bold**. Second best in underline.

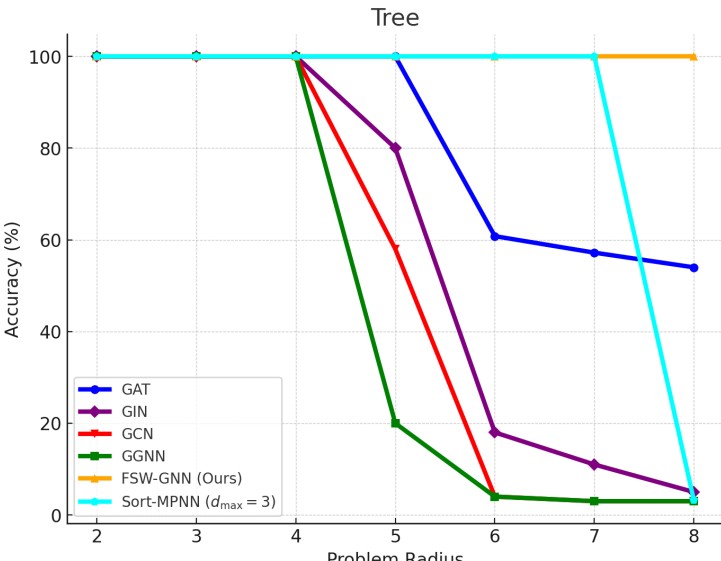

Figure 2: Trees models comparison.

consider the 'graph transfer' tasks from Di Giovanni et al. (2023). In this problem, a feature from a 'source' node is propagated to a 'target' node, which is $r$ message passing steps away from the source node. All other nodes have 'blank' node features. We consider this problem for the three different graph topologies suggested by Di Giovanni et al. (2023): clique, ring, and crossring, and with a problem radius $r$ varying from 2 to 15.

As shown in Figure 3, FSW-GNN is the only method attaining $100\%$ accuracy across all three graphs and radii. Other models start failing at much smaller $r$, where the value of this $r$ depends on the graph topology. We do find that our model performance does deteriorate when $r > 20$.

Finally, we evaluate the possible relationship of our empirical results with *oversmoothing*. Over-smoothing is a phenomenon that often occurs with deep MPNNs, where all node features become nearly identical as depth increases, thus leading to degraded performance. Oversmoothing is often measured using the Dirichlet energy or Mean Average Distance (MAD) (see Rusch et al. (2023) for the formulas), which becomes closer to zero when the feature vectors are closer to each other. In Figure 4 we measure the MAD metric of all methods on the Ring experiment. The figure shows that all methods except for FSW-GNN exhibit over-smoothing starting from some $r$ (that is, have a near 0 MAD energy), and this correlates pretty well with the performance of the methods in terms of accuracy, shown in Figure 3 (middle).

We note that long-range issues can be alleviated using graph rewiring methods which reduce the radius of the problem Gutteridge et al. (2023), or add global information using spectral filters Geisler et al. (2024) or graph transformers. What is special about FSW-GNN is that it performs well with the given graph topology. Thus, this work gives us a direction to address the core problem of relaying long-range messages efficiently rather than using methods to circumvent the problem.

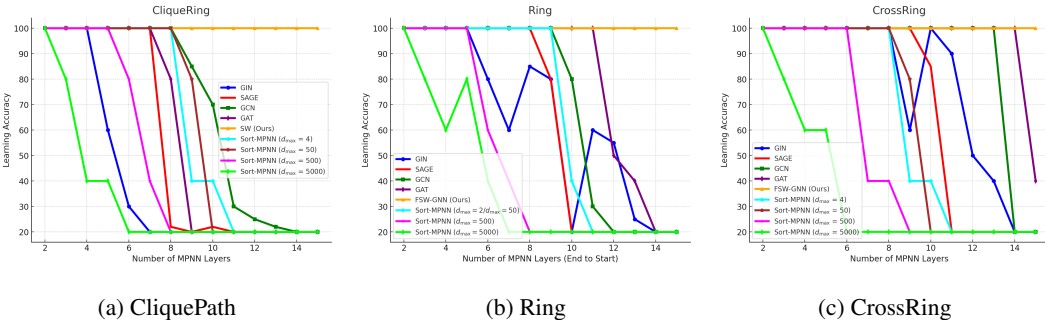

(a) CliquePath  (b) Ring  (c) CrossRing

Figure 3: Performance comparison of MPNN models across the CliquePath, Ring, and CrossRing graph transfer tasks as presented in Di Giovanni et al. (2023).

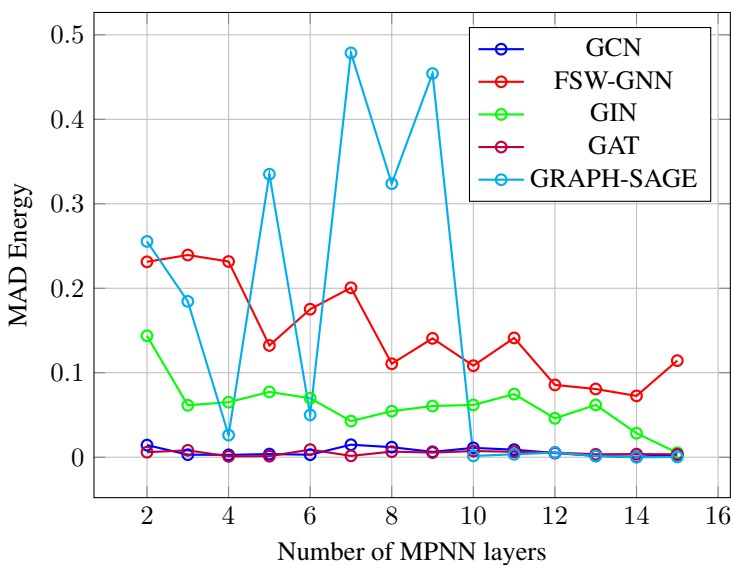

Figure 4: Dirichlet Energy vs. Number of MPNN layers for various models on the Ring long-range task

## 5 CONCLUSION

In this paper, we introduced FSW-GNN, the first bi-Lipschitz MPNN. Empirically, we have found that FSW-GNN is very effective for long range learning problems. Our current explanation for why this should be the case is discussed at the end of Section 2. Our main goal for future work is to strengthen and formalize this explanation. In particular, currently we have no control on the bi-Lipschitz distortion of FSW-GNN, and its dependence on depth. Informally, we would expect that if each FSW embedding has distortion of $C/c$, then the total FSW-GNN will be $(C/c)^T$, growing exponentially with the depth $T$. However, the stability of FSW-GNN for problems of rather large radius indicate that the rate of distortion growth is much smaller. A possible explanation for this is that, at the limit where the width goes to infinity, FSW embeddings have a optimal distortion of 1 with respect to the sliced Wasserstein distance. Accordingly, in future work we will aim to understand how to control the distortion of FSW-GNN, and formally prove the low distortion FSW-GNN, with respect to an appropriate node-WL-metric, avoids both oversmoothing and oversquashing.

A limitaton of FSW-GNN is that, due to its more complex aggregation, its runtime is higher than standard MPNN: for the LRGB struct the average time per epoch of FSW-GNN is four times slower than GIN and GCN, as shown in Appendix Table 6.

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

## A   STATISTICS ON OUR BENCHMARKS

In the first table, we present each transductive dataset and its statistics for the MUTAG dataset, including the number of nodes, edges, features, classes, average degree, and density, measuring the number of edges divided by the number of maximal edges. As we can see in 4, those datasets are very sparse.

Next, we add the same table for the LRGB tasks.

| Dataset | Cora | Cite. | Pubm. | Cham. | Squi. | Actor | Corn. | Texa. | Wisc. | MUTAG |
|---|---|---|---|---|---|---|---|---|---|---|
| # Nodes | 2708 | 3327 | 19717 | 2277 | 5201 | 7600 | 183 | 183 | 251 | 188 |
| # Edges | 5429 | 4732 | 44338 | 36101 | 217073 | 33544 | 295 | 309 | 499 | 744 |
| # Features | 1433 | 3703 | 500 | 2325 | 2089 | 931 | 1703 | 1703 | 1703 | 7 |
| # Classes | 7 | 6 | 3 | 5 | 5 | 5 | 5 | 5 | 5 | 2 |
| **Avg. Degree** | 4 | 2 | 4 | 31 | 83 | 8 | 3 | 3 | 4 | 8 |
| **Density** | 0.0007 | 0.0009 | 0.0001 | 0.0159 | 0.0161 | 0.0012 | 0.0177 | 0.0184 | 0.0041 | 0.0422 |

Table 4: Graph statistics for common datasets.

| Dataset | Peptides-Func | Peptides-Struct |
|---|---|---|
| # Graphs | 15,535 | 15,535 |
| # Nodes (Avg.) | 150.94 | 150.94 |
| # Edges (Avg.) | 2.04 | 2.04 |
| **Avg. Degree** | 2.04 | 2.04 |
| **Density** | $1.74 \times 10^{-6}$ | $1.74 \times 10^{-6}$ |
| # Classes | - | 10 |

Table 5: Graph statistics for Peptides datasets.

## B   RELATION TO TREE MOVER'S DISTANCE

The Tree Mover's Distance comes from Chuang & Jegelka (2022), who showed that it is equivalent to WL. This metric is based on a tree distance between computation trees that simulate the WL test. It is defined recursively, roughly as the sum of the distance between tree roots $r$ and a Wasserstein distance between the sub-trees rooted at $r$'s children. For WL equivalence, this metric assumes the feature domain $\Omega$ does not contain the zero vector.

We first review Wasserstein distances. Recall that if $(\mathbf{X}, d)$ is a metric space, $\Omega \subseteq X$ is a subset, then the Wasserstein distance can be defined on the space of multisets consisting of $n$ elements in $\Omega$ via

$$W_1(x_1, \ldots, x_n, y_1, \ldots, y_n) = \min_{\tau \in S_n} \sum_{j=1}^{n} d(x_j, y_{\tau(j)})$$

For multisets of different size, the authors of (Chuang & Jegelka, 2022) used an augmentation map, which, for a fixed parameter $n$, augments multisets of size $r \leq n$ by padding with $n - r$ instances of the zero vector:

$$\Gamma(x_1, \ldots, x_r) = (x_1, \ldots, x_r, x_{r+1} = 0, \ldots x_n = 0)$$

and the augmented distance on multi-sets of size up to $n$ is defined by

$$W_d(X, \hat{X}) = W_1\Big(\Gamma(X), \Gamma\big(\hat{X}\big)\Big).$$

We now return to define the TMD. We consider the space of graphs $\mathcal{G}_{\leq N}(\Omega)$, consisting of graphs with $\leq N$ nodes, with node features coming from a compact domain $\Omega \subseteq \mathbb{R}^d$ such that $0 \notin \Omega$. The TMD is defined using the notion of computation trees:

**Definition B.1.** (Computation Trees). Given a graph $\mathcal{G} = (V, E, X)$ with node features $\{x_v\}_{v \in V}$, let $T_v^1$ be the rooted tree with a single node $v$, which is also the root of the tree, and node features $x_v$. For $K \in \mathbb{N}$ let $T_v^K$ be the depth-$K$ computation tree of node $v$ constructed by connecting the neighbors of the leaf nodes of $T_v^{K-1}$ to the tree. Each node is assigned the same node feature it had in the original graph $\mathcal{G}$. The multiset of depth-$K$ computation trees defined by $\mathcal{G}$ is denoted by $\mathcal{T}_G^K := \{T_v^K\}_{v \in V}$. Additionally, for a tree $T$ with root $r$, we denote by $\mathcal{T}_r$ the multiset of subtrees that root at the descendants of $r$.

**Definition B.2.** (Blank Tree). A blank tree $\bar{T}_0$ is a tree (graph) that contains a single node and no edge, where the node feature is the zero vector $0$.

Recall that by assumption, all node features will come from the compact set $\Omega$, and $0 \notin \Omega$.

We can now define the tree distance:

**Definition B.3.** (Tree Distance).[2] The distance between two trees $T_a, T_b$ with features from $\Omega$ and $0 \notin \Omega$, is defined recursively as

$$TD(T_a, T_b) := \begin{cases} \|x_{r_a} - x_{r_b}\|_1 + W_{TD}^{\bar{T}_0}(\mathcal{T}_{r_a}, \mathcal{T}_{r_b}) & \text{if } K > 1 \\ \|x_{r_a} - x_{r_b}\|_1 & \text{otherwise} \end{cases}$$

where $K$ denotes the maximal depth of the trees $T_a$ and $T_b$.

**Definition B.4.** (Tree Mover's Distance). Given two graphs, $G_a, G_b$ and $w, K \geq 0$, the tree mover's distance is defined by

$$TMD^K(G_a, G_b) = W_{TD}^{\bar{T}_0}(\mathcal{T}_{G_a}^K, \mathcal{T}_{G_b}^K),$$

where $\mathcal{T}_{G_a}^K$ and $\mathcal{T}_{G_b}^K$ denote the multiset of all depth $K$ computational trees arising from the graphs $G_a$ and $G_b$, respectively. Chuang & Jegelka (2022) proved that $TMD^K(G_a, G_b)$ is a pseudo-metric that fails to distinguish only graphs that cannot be separated by $K+1$ iterations of the WL test. Thus, assuming that $0 \notin \Omega$, $TMD^K(G_a, G_b)$ is WL equivalent on $\mathcal{G}_{\leq N}(\Omega)$.

In addition, it is easy to see from the definition of TMD that it satisfies the following properties:

1. $TMD^K((\mathbf{A}, \alpha \cdot \mathbf{X}), (\mathbf{B}, \alpha \cdot \mathbf{Y})) = \alpha \cdot TMD^K(\mathbf{X}, \mathbf{Y})$ for any $\alpha \geq 0$.

2. The TMD metric is piecewise linear in $(\mathbf{X}, \mathbf{Y})$.

These properties will be used to show that under the above assumptions, the embedding computed by FSW-GNN is bi-Lipschitz with respect to TMD.

## C PROOFS

### C.1 DS METRIC

Here we prove Theorem 3.1 that says that our augmented DS metric, defined in (8) is indeed a WL metric.

**Theorem 3.1.** [Proof in Appendix C.1] *Let $\rho_{\text{DS}} : \mathcal{G}_{\leq N}(\mathbb{R}^d) \times \mathcal{G}_{\leq N}(\mathbb{R}^d) \to \mathbb{R}_{\geq 0}$ be as in (8). Then $\rho_{\text{DS}}$ is a WL-equivalent metric on $\mathcal{G}_{\leq N}(\mathbb{R}^d)$.*

*Proof.* We first prove that it is symmetric and satisfies the triangle inequality. The metric is symmetric as if a pair's minimum is obtained at $S$ then the exact value is obtained for the opposite pair with $S^T$ and vice-versa. We point out that the matrix norm we consider is a sub-multiplicative norm, like the operator norm, Forbinius norm or the $l1$ norm. Let $(V_1, \mathbf{A}, \mathbf{X}), (V_2, \mathbf{B}, \mathbf{Y}), (V_3, \mathcal{C}, \mathbf{Z})$ three

---

[2]This definition slightly varies from from the original definition in Chuang & Jegelka (2022), due to our choice to set the depth weight to 1 and using the 1-Wasserstein which is equivalent to optimal transport.

arbitrary graphs with $|V_i| = n_i$, $i = 1, 2, 3$. Then the following holds:

$$\alpha := d_{\mathbf{A},\mathbf{B}} = |n_1 - n_2| + min_{S \in \Pi(n_1,n_2)}||\mathbf{A}S - S\mathbf{B}|| + \sum_{i,j} S_{i,j}||\mathbf{X}_i - \mathbf{Y}_j||$$

$$\beta := d_{\mathbf{B},\mathcal{C}} = |n_2 - n_3| + min_{S \in \Pi(n_2,n_3)}||\mathbf{B}S - S\mathcal{C}|| + \sum_{i,j} S_{i,j}||\mathbf{Y}_i - \mathbf{Z}_j||$$

$$\gamma := d_{\mathbf{A},\mathcal{C}} = |n_1 - n_3| + min_{S \in \Pi(n_1,n_3)}||\mathbf{A}S - S\mathcal{C}|| + \sum_{i,j} S_{i,j}||\mathbf{X}_i - \mathbf{Z}_j||$$

We want to prove $\gamma \leq \beta + \alpha$. The minimum of the first two equations is obtained in $S^1, S^2$. Define $S^3 = S^1 \cdot S^2$ and note that $S^3 \in \Pi(n_1, n_3)$. We use the property that for $S \in \Pi(n_1, n_2)$, $T \in \Pi(n_2, n_3)$, $||S \cdot T|| \leq ||S|| \cdot ||T||$, and $||S|| \leq 1$. This holds, for example, for entrywise $\ell_p$-norms for $p \geq 1$ and for the $\ell_2 - \ell_2$ operator norm. Then

$$||\mathbf{A}S^3 - S^3\mathcal{C}|| = ||\mathbf{A}S^1 S^2 - S^1 S^2\mathcal{C}|| = ||\mathbf{A}S^1 S^2 - S^1 \mathbf{B}S^2 + S^1 \mathbf{B}S^2 - S^1 S^2\mathcal{C}||$$

$$\leq ||\mathbf{A}S^1 S^2 - S^1 \mathbf{B}S^2|| + ||S^1 \mathbf{B}S^2 - S^1 S^2\mathcal{C}|| = ||(\mathbf{A}S^1 - S^1 B) \cdot S^2|| + ||S^1 \cdot (\mathbf{B}S^2 - S^2\mathcal{C})||$$

$$\leq ||S^2|| \cdot ||\mathbf{A}S^1 - S^1 \mathbf{B}|| + ||S^1|| \cdot ||\mathbf{B}S^2 - S^2\mathcal{C}|| \leq ||\mathbf{A}S^1 - S^1 \mathbf{B}|| + ||\mathbf{B}S^2 - S^2\mathcal{C}||$$

Next, we show the second part is also smaller:

$$\sum_{i,j} S^3_{i,j} \cdot |X_i - Y_j| = \sum_{i,j} \sum_k S^1_{i,k} \cdot S^2_{k,j} \cdot ||X_i - Z_j|| \leq \sum_{i,j} \sum_k S^1_{i,k} \cdot S^2_{k,j} \cdot (||X_i - Z_k|| + ||Z_k - Y_j||)$$

$$= \sum_{i,j} \sum_{k=1}^n S^1_{i,k} \cdot S^2_{k,j} \cdot ||X_i - Z_k|| + \sum_{i,j} \sum_{k=1}^n S^1_{i,k} \cdot S^2_{k,j} \cdot ||Y_j - Z_k||$$

We now open both sums:

$$\sum_{i,j} \sum_k S^1_{i,k} \cdot S^2_{k,j} \cdot ||\mathbf{X}_i - \mathbf{Z}_k|| = \sum_k \sum_j \sum_i S^1_{i,k} \cdot S^2_{k,j} \cdot ||\mathbf{X}_i - \mathbf{Z}_k|| = \sum_k \sum_j S^2_{k,j} \sum_i S^1_{i,k} \cdot ||\mathbf{X}_i - \mathbf{Z}_k|| =$$

$$\sum_k \sum_j S^2_{k,j} \cdot f_k = \sum_k f_k \sum_j S^2_{k,j} = \sum_k f_k = \sum_{i,j} S^1_{i,j} \cdot ||\mathbf{X}_i - \mathbf{Y}_j||$$

With the same argument, we obtain that

$$\sum_{i,j} \sum_k S^1_{i,k} \cdot S^2_{k,j} \cdot ||\mathbf{Y}_j - \mathbf{Z}_k|| = \sum_{i,j} S^2_{i,j} \cdot ||\mathbf{Y}_i - \mathbf{Z}_j||$$

So overall we found matrix $S^3$ such that:

$$||\mathbf{A}S^3 - S^3\mathcal{C}|| + \sum_{i,j} S^3_{i,j}||\mathbf{X}_i - \mathbf{Z}_j|| \leq ||\mathbf{A}S^1 - S^1\mathbf{B}|| + \sum_{i,j} S^1_{i,j}||\mathbf{X}_i - \mathbf{Y}_j|| + ||\mathbf{B}S^2 - S^2\mathcal{C}||$$

$$+ \sum_{i,j} S^3_{i,j}||\mathbf{Y}_i - \mathbf{Z}_j|| = \alpha + \beta$$

We took specific feasible matrices in the minimization problems, and thus the minimum is even smaller, so $\gamma \leq \alpha + \beta$.

Now, we show that our metric $\rho_{\text{DS}}$ is equivalent to WL. Clearly any pair of graphs with different numbers of vertices are distinguished both by WL and by $\rho_{\text{DS}}$. Thus, in the following we assume that the two graphs have the same number of vertices.

We begin first with some needed definitions.

**Partitions** Most of our techniques are inspired by Scheinerman & Ullman (2013). Given $\mathcal{G} = (\mathbf{A}, \mathbf{X})$, we define a stable partition $\mathcal{V} = \mathcal{P}_1 \cup \mathcal{P}_2 ... \cup \mathcal{P}_k$ if

$$\forall u, v \in \mathcal{P}_i, \mathbf{X}_u = \mathbf{X}_v \tag{15}$$

$$\forall l \in [k], |\mathcal{N}_u \cap \mathcal{P}_l| = |\mathcal{N}_v \cap \mathcal{P}_l| \tag{16}$$

In simple words, two nodes in the same partition must have the same feature and the same number of neighbors with the same feature. Note that each singleton is a valid stable partition. We can characterize a partition as a $k$ tuple, and in the $i'th$ place, we put a tuple of the common feature and a vector telling the number of neighbors in other partitions. We say that graphs have the same stable partition; if, up to permitting of the $k$ tuple indices, we have the same $k$ tuple.

**Lemma C.1.** *The number of colors in 1-WL can't increase at level $n$. In addition, all nodes with the same color at step $n$ make a stable partition.*

*Proof.* By the pinhole principle, as the number of colors can't decrease, after at most $n$ iterations, the number of different colors will stay the same. Denote by $\mathcal{P} = (\mathcal{P}_1, .., \mathcal{P}_s)$ a partition of the nodes with the same coloring at the $n$ iteration, and be $c \in C$ some color, we claim that nodes in the same partition, have the same number of neighbors with color $c$. As otherwise those nodes that now have same coloring will have different coloring in the next $n + 1$ iteration. But as the number of colors doesn't decrease (because of the concatenation of the current color), we have at least one more color, a contradiction. So, we found a stable partition. $\square$

**Lemma C.2.** *The following conditions are equivalent:*

- *$\mathcal{G} \cong_{1-WL} \mathcal{H}$*

- *Both graphs have a common stable partition.*

*Proof.* Assume we have the same common partition. Up to renaming the names of the vertices of the nodes in the first graphs, we assume we have the same stable partition with the same parameters. Assume that a common stable partition exists $\mathcal{P} = \mathcal{P}_1 \cup .. \cup P_k$. Then, by simple induction on the number of iterations, we will prove that all nodes in the same partition have the same color in both graphs.
**Basis** Nodes in the same partition have the same initial feature and, thus, have the same color (in both graphs).
**Step** By the induction hypothesis, all nodes in the same partition iteration $T$ have the same color; and they both have the same number of neighbors with the same color, so the aggregation yields the same output. Thus, in iteration $T + 1$, those nodes also have the same color. Note that this argument is symmetric to both graphs; thus, the 1-WL test will not distinguish them.

On the other hand, if we have the same 1-WL embedding, then we partition the nodes to those classes with the same color at iteration $n$. We have to show this partition is valid. First, by definition, those nodes $u, v \in \mathcal{P}_i$ have the same node feature (by iteration 1); next, if $u, v$ don't have the same combinatorial degree in $\mathcal{P}_l$, then their color won't be the same at iteration $n + 1$. But, as shown in lemma C.1, the number of distinct colors can't increase at iteration $n$. Thus, we found a common stable partition. $\square$

Before proving the following lemma, we revise a definition from Scheinerman & Ullman (2013). Given $S \in \mathbb{R}^{n \times n}$, we say $S$ is composable if there exists $P, Q, S_1, S_2$ such that

$$S = P \cdot (S_1 \oplus S_2) \cdot Q$$

Such that $P, Q$ are permutation matrices. By simple induction, we can write $M$ as a direct sum of an indecomposable

$$S = P \cdot (S_1 \oplus S_2 \oplus ... \oplus S_t) \cdot Q$$

**Lemma C.3.** *Let $S \in \Pi(n, n)$ and assume it's in the form of $S = S_1 \oplus ... \oplus S_k$ such that all blocks are indecomposable and $\sum_{i,j} S_{i,j} \cdot ||\mathbf{X}_i - \mathbf{Y}_j|| = 0$. Denote by $i_1, ..., i_{k+1}$ the indices define the start and the end of $S_1, ..., S_t$ and by $I_k := [i_k, i_{k+1}]$. Then $\mathbf{X}_i = \mathbf{Y}_j, \forall t \in [k], \forall i, j \in I_t$.*

*Proof.* Given $t \in [k]$, we build a bipartite graph from $I_t$ to itself, such that two indexes $i, j$ are connected if $S_{i,j} > 0$. Note that this graph is connected, as otherwise, $S_t$ could be composed into its connected components, and thus $S_t$ would be composable. Given a path $P = (i_1, ..., i_l)$ in the graph, note that by definition, as the metric vanishes, and $S_{i_j, i_{j+1}} > 0$, so $\mathbf{X}_{i_j} = \mathbf{Y}_{i_{j+1}}$, thus $\mathbf{X}_i = \mathbf{Y}_j, i \in I_k, j \in I_k$ and we are done. $\square$

**Theorem C.4.** *Be $\mathcal{G}, \mathcal{H}$ two featured graphs, then*

$$d(\mathcal{G}, \mathcal{H}) = 0 \iff \mathcal{G} \overset{\text{WL}}{\sim} \mathcal{H}$$

We first prove that if the two graphs are 1-WL equivalent, then this metric vanishes. We may assume they have the same stable partition as we proved above. Take $\mathcal{P} = \mathcal{P}_1 \cup .. \cup \mathcal{P}_t$, rename the nodes such that they come in consecutive order and denote by $n_1, n_2, .., n_k$ the sizes of the partitions. Note that by definition, $\forall u, v \in \mathcal{P}_i$, both have the same feature and number of neighbors in all $\mathcal{P}_l$. As in the book Scheinerman & Ullman (2013), define $S := \frac{1}{n_1} J_{n_1} \oplus ... \oplus \frac{1}{n_j} J_{n_k}$ and from the book Scheinerman & Ullman (2013), we know that $\mathbf{A}S = S\mathbf{B}$. As all nodes in the same partition have the same feature, $\mathbf{X}_i = \mathbf{Y}_j, \forall i, j \in I_k = [n_t, n_{t+1}], \forall t \in [k]$, so as $S$ is non-zero only on indexes in the same partition then also this metric vanishes, so also the sum vanishes on $S$.

For the next direction, be $\mathcal{G} = (\mathbf{A}, \mathbf{X}), \mathcal{H} = (\mathbf{B}, \mathbf{Y})$ two graphs and assume there exists a matrix of the form of $S = P(S^1 \oplus ... \oplus S_t)Q$ such that the metric vanishes and denote by $D = S^1 \oplus ... \oplus S_t$. We show we can choose $S$ such that it's a block matrix.

$$||PDQ \cdot \mathbf{A} - \mathbf{B} \cdot PDQ|| = ||P \cdot (DQ\mathbf{A} \cdot Q^{-1} - P^{-1}\mathbf{B}P \cdot D) \cdot Q|| =_* ||DQ\mathbf{A} \cdot Q^{-1} - P^{-1}\mathbf{B}P \cdot D||$$

$$\sum_{i,j} S_{i,j} \cdot |\mathbf{X}_i - \mathbf{Y}_j| = \sum_{i,j} (PDQ)_{i,j} \cdot |\mathbf{X}_i - \mathbf{Y}_j| = \sum_{i,j} D_{\pi_1^{-1}(i), \pi_2(j)} \cdot |\mathbf{X}_i - \mathbf{Y}_j| =$$

$$\sum_{i,j} D_{i,j} \cdot |\mathbf{X}_{\pi_1(i)} - \mathbf{Y}_{\pi_2^{-1}(j)}|$$

(*) - note that the first equality is because permutation matrices preserve the norm. We define two graphs $\hat{\mathcal{G}} = (Q\mathbf{A}Q^{-1}, Q\mathbf{X}) \cong \mathcal{G}, \hat{\mathcal{H}} = (P^{-1}\mathbf{A}P, P^{-1}\mathbf{Y}) \cong \mathcal{H}$. So, we can choose $S$ to be a diagonal block matrix. Note that $D$ defines a partition of the nodes, and we will prove that it's a stable partition of both graphs. From $SA = BS$, we obtain, as in the book Scheinerman & Ullman (2013), that each of the two nodes $u, v \in \mathcal{P}_k$ has the same number of neighbors in $\mathcal{P}_l, \forall l \in [t]$. Next, by the lemma C.3, we know that $\mathbf{X}_i = \mathbf{Y}_j, \forall i, j \in I_k$, so nodes in the same partition have the same feature. Thus, this partition is stable. So, $\hat{\mathcal{G}}, \hat{\mathcal{H}}$ have exactly the same partition. Then $\mathcal{G}, \mathcal{H}$ have the same partition up to isomorphism, and by lemma C.2, both graphs are 1-WL equivalent. $\square$

## C.2 FSW-GNN: Equivalence to WL

We now prove Theorem 3.2 that says the FSW-GNN is equivalent to WL.

**Theorem 3.2** (Informal)**.** [Proof in Appendix C.2] *Consider the FSW-GNN architecture for input graphs in $\mathcal{G}_{\leq N}(\mathbb{R}^d)$, with $T = N$ iterations, where $\Phi^{(t)}, \Psi$ are just linear funtions, and all features (except for input features) are of dimension $m \geq 2Nd + 2$. Then for Lebesgue almost every choice of model parameters, the graph embedding defined by the architecture is WL equivalent.*

The formal requirements of the theorem are as follows: $\Phi^{(t)}$ and $\Psi$ are all matrices with output dimension $m$, whose entries are drawn independently from a absolutely-continuous distributions over $\mathbb{R}$. The FSW embeddings depend on random parameters as described in (Amir & Dym, 2024). These embeddings should be generated independently, and have output dimension $m$.

Under these assumptions, with probability 1, all the matrices $\Phi^{(t)}$ and $\Psi$ and FSW embedding instances are injective (see in (Amir & Dym, 2024) Theorem 4.1 and Appendix A.1), and therefore the resulting MPNN is WL equivalent.

## C.3 FSW-GNN: Bi-Lipschitzness

We first prove Lemma 3.4.

**Lemma 3.4.** [Proof in Appendix C.3] *Let $f, g : M \to \mathbb{R}_{\geq 0}$ be nonnegative piecewise-linear functions defined on a compact polygon $M \subset \mathbb{R}^d$. Suppose that for all $x \in M$, $f(x) = 0$ if and only if $g(x) = 0$. Then there exist real constants $c, C > 0$ such that*

$$c \cdot g(x) \leq f(x) \leq C \cdot g(x), \quad \forall x \in M. \tag{14}$$

*Proof.* It is enough to prove the left-hand side of (14). The right-hand side can then be proved by reversing the roles of $f$ and $g$.

Let $A = \{x \in M \mid g(x) > 0\}$. Suppose by contradiction that there exists a sequence $\{x_i\}_{i=1}^{\infty} \in A$ such that

$$\frac{f(x_i)}{g(x_i)} \xrightarrow[i \to \infty]{} 0. \tag{17}$$

Since $M$ is compact, we can assume without loss of generality that $x_i \xrightarrow[i \to \infty]{} x_0 \in M$. Since $g(x)$ is bounded on $M$, equation (17) implies that $f(x_i) \xrightarrow[i \to \infty]{} 0$. By continuity, $f(x_0) = 0$ and thus $g(x_0) = 0$.

Let $L_1, \ldots, L_K \subseteq M$ be the mutual refinement of the linear regions of $f$ and $g$. That is, on each $L_k$, both $f$ and $g$ are linear. Moreover, by a finite number of further refinements, we can assume that all the $L_k$'s are compact convex polytopes. Since there are finitely many $L_k$'s, by taking a subsequence of $\{x_i\}_{i=1}^{\infty}$, we can assume without loss of generality that all of the $x_i$'s belong to $L_1$. Since $L_1$ is compact, $x_0 \in L_1$.

Since $f(x_0) = g(x_0) = 0$, the restriction of $f$ and $g$ to $L_1$ can be expressed by

$$f(x) = \langle a, x - x_0 \rangle, \quad g(x) = \langle b, x - x_0 \rangle, \tag{18}$$

with $a, b \in \mathbb{R}^d$ being constant vectors.

Let $v_1, \ldots, v_r$ be the vertices of the convex polytope $L_1$, and let $K \subseteq \mathbb{R}^d$ be the polyhedral cone generated by $\{v_1 - x_0, \ldots, v_r - x_0\}$, namely

$$K := \left\{ \sum_{i=1}^{r} \theta_i (v_i - x_0) \ \middle| \ \theta_1, \ldots, \theta_r \geq 0 \right\}. \tag{19}$$

First, note that by construction, $x_0 + K \supseteq L_1$. To see this, set one $\theta_i$ to 1 and the rest to zero in (19). This yields $x_0 + K \ni x_0 + 1(v_i - x_0) = v_i$. Since $x_0 + K$ contains all the vertices of $L_1$ and both sets are convex, $x_0 + K$ contains $L_1$.

Second, we argue that any $x \in x_0 + K$ that is sufficiently close to $x_0$ belongs to $L_1$. This can be shown by noting that if the distance from $x \in x_0 + K$ to $x_0$ is small enough, it can be written as $x = x_0 + \sum_{i=1}^{r} \theta_i (v_i - x_0)$ with $\sum_{i=1}^{r} \theta_i \leq 1$. Set $\theta_0 = 1 - \sum_{i=1}^{r} \theta_i$. Then

$$x = \theta_0 x_0 + \sum_{i=1}^{r} \theta_i v_i,$$

and since $x$ is a convex combination of points that belong to $L_1$ it is in $L_1$.

Thus:

$$\begin{aligned}
0 = \lim_{i \to \infty} \frac{f(x_i)}{g(x_i)} &\geq \inf_{x \in L_1 \cap A} \frac{f(x)}{g(x)} = \inf_{x \in L_1 \cap A} \frac{\langle a, x - x_0 \rangle}{\langle b, x - x_0 \rangle} \\
&\overset{(a)}{=} \inf_{u \in (L_1 - x_0) \cap (A - x_0)} \frac{\langle a, u \rangle}{\langle b, u \rangle} \overset{(b)}{=} \inf_{u \in (L_1 - x_0) \mid \langle b, u \rangle > 0} \frac{\langle a, u \rangle}{\langle b, u \rangle} \\
&\overset{(c)}{\geq} \inf_{u \in K \mid \langle b, u \rangle > 0} \frac{\langle a, u \rangle}{\langle b, u \rangle} \overset{(d)}{=} \inf_{u \in K \mid \langle b, u \rangle = 1} \langle a, u \rangle,
\end{aligned} \tag{20}$$

where (a) is by change of variables $u = x - x_0$; (b) is since the domains $(L_1 - x_0) \cap (A - x_0)$ and $\{u \in L_1 - x_0 \mid \langle b, u \rangle > 0\}$ are identical; (c) holds since $K \supseteq L_1 - x_0$; and (d) holds since $K$ is closed to positive scalar multiplication, and the ratio $\frac{\langle a, u \rangle}{\langle b, u \rangle}$ is invariant to such scaling.

Let $B = \{u \in K \mid \langle b, u \rangle = 1\}$. It follows from (20) that either there exists $u_0 \in B$ such that $\langle a, u_0 \rangle \leq 0$, or

$$\inf_{u \in B} \langle a, u \rangle = 0 \tag{21}$$

and the infimum is not attained. We shall now show that the latter option is impossible. Note that for all $u \in \mathbb{R}^d$,

$$|\langle a, u \rangle| = \|a\| \cdot \text{dist}(u, a^{\perp}),$$

where $a^\perp$ is the hyperplane in $\mathbb{R}^d$ perpendicular to $a$, and $\text{dist}(u, a^\perp)$ is the Euclidean distance from $u$ to the set $a^\perp$. Since the infimum in (21) equals zero, the distance between $B$ and $a^\perp$ equals zero. Although not compact, both $B$ and $a^\perp$ are closed convex polytopes, and thus $\text{dist}(B, a^\perp) = 0$ implies that the two sets intersect.[3] Hence, the infimum of zero in (21) is attained, which is a contradiction. Thus, there exists $u_0 \in B$ such that $\langle a, u_0 \rangle \leq 0$.

Let $\delta > 0$ small enough such that $x_0 + \delta u_0 \in L_1$, and thus

$$0 \leq f(x_0 + \delta u_0) = \langle a, (x_0 + \delta u_0) - x_0 \rangle = \delta \langle a, u_0 \rangle \leq 0,$$

and

$$g(x_0 + \delta u_0) = \langle b, (x_0 + \delta u_0) - x_0 \rangle = \delta \langle b, u_0 \rangle = \delta \cdot 1 > 0,$$

which is a contradiction. $\square$

We now turn to the full proof of Theorem 3.3.

**Theorem 3.3.** [Proof in Appendix C.3] *Let $\Omega \subset \mathbb{R}^d$ be compact. Under the assumptions of Theorem 3.2, the FSW-GNN is bi-Lipschitz with respect to $\rho_{\text{DS}}$ on $\mathcal{G}_{\leq N}(\Omega)$. If, additionally, $\Omega$ is a compact polygon that does not contain $\mathbf{0}$, then the FSW-GNN is bi-Lipschitz with respect to TMD on $\mathcal{G}_{\leq N}(\Omega)$.*

*Proof.* Following the reasoning in the main text, the rest of the proof is as follows: Let $G, \tilde{G} \in \mathcal{G}_{\leq N}(\Omega)$, $G = (V, E, X)$ and $\tilde{G} = (\tilde{V}, \tilde{E}, \tilde{X})$, with $n$ and $\tilde{n}$ vertices respectively. It is enough to show that the bi-Lipschitz ratio is bounded on all choices of $X \in \Omega^n$, $\tilde{X} \in \Omega^{\tilde{n}}$, since there is a finite number of choices of $n, \tilde{n} \leq N$ and edges $E, \tilde{E}$. Define the function $f(X, \tilde{X}) = \|\mathbf{h}_G - \mathbf{h}_{\tilde{G}}\|_1$, $g(X, \tilde{X}) = \rho(G, \tilde{G})$, where $\rho$ is either $\rho_{\text{DS}}$ or TMD. By the comment above, both $f$ and $g$ are piecewise linear. Since both FSW-GNN and the metric $\rho$ are WL-equivalent, $f$ and $g$ have the same zero set. Thus, Lemma 3.4 guarantees the existence of Lipschitz constants $c, C$.

Finally, note that by equivalence of all norms on a finite dimensional space, the same bi-Lipschitz equivalence holds if we replace the one-norm in the definition of $f$ with any other norm. Q.E.D. $\square$

# D  EXPERIMENT DETAILS

We used the Adam optimizer for all experiments.

For the empirical distortion evaluation, we used pairs of graphs $G, \tilde{G}$, each of which consisting of four vertices and the edges $1-2-3-4-1$. Two random vectors $v_0, \Delta v \in \mathbb{R}^d$ were drawn i.i.d. Gaussian and normalized to unit length. In $G$ all vertex features were set to $v_0$, whereas in $\tilde{G}$ they were set to $v_0 + \varepsilon \sigma \Delta v$, with $\sigma = 1$ for $v_2, v_4$ and $-1$ for $v_1, v_3$. We used $\varepsilon = \{1, 1e-1, \ldots, 1e-6\}$, and generated 100 pairs for each value of $\varepsilon$, and evaluated the constants $C, c$ for the resulting pairs. This experiment was repeated 10 times and the average distortion was taken. To ensure accurate results, we used 64-bit floating-point arithmetic in this experiment.

For the NeighborsMatch problem from (Alon & Yahav, 2020), we used the protocol developed in this paper: we used their implementation for the MPNNs we compared to, with a hidden dimension of 64 for all models, searched for each of its best hyper-parameters, and reported the training accuracy. For fair comparison with rival models, we repeated each sample 100 times, as was done in Alon & Yahav (2020).

For the Ring dataset, we used the results from Di Giovanni et al. (2023) and trained our models with a hidden dimension of 64.

For the LRGB dataset, we trained all models under the constraint of 500K parameters. In contrast, for the MolHIV dataset, there was no restriction, and we trained the models with 40K parameters.

For the transductive learning tasks, we used a hidden dimension of 128 across all models.

---

[3]This holds since the distance between two closed convex polytopes can be presented as the optimal objective value of a *quadratic program* (QP), which always has an optimal solution; see, for example, Part Proposition 1.4.12 in (Bertsekas, 2009).

**Timing**  Here we show the average time per epoch for FSW-GNN, GIN and GCN, on the Peptides Struct task.

| Model | GIN | GCN | FSW-GNN |
|---|---|---|---|
| Avg Time per Epoch | 12.16 | 14.25 | 49.3 |

Table 6: Comparison of Average Time per Epoch for Different Models

