# OpenReview forum: "FSW-GNN: A Bi-Lipschitz WL-Equivalent Graph Neural Network"
_ICLR.cc/2025/Conference — Submitted to ICLR 2025_

### Official Review · Reviewer_yDYZ · 2024-11-01

**Soundness:** 2
**Presentation:** 2
**Contribution:** 2
**Rating:** 5
**Confidence:** 5

**Summary:**

Motivated by recent observations that the quality of separation by common Weisfeiler-Leman (WL) expressive message passing graph neural networks (MPNNs) can be quite low, this work presents an MPNN architecture which is -- in contrast to standard architectural choices -- WL-equivalent. This means that (almost) *all* functions in the function class can distinguish *all* pairs of WL-distinguishable graphs. The new architecture, called FSW-GNN, follows the standard MPNN framework, with the essential difference being that multiset functions used for neighborhood aggregation are replaced by the ~~(almost)~~ injective Fourier-Sliced Wasserstein (FSW) embedding for multisets. Further, Lipschitzness of this architecture is analyzed w.r.t. two WL distances, the tree mover's distance (TMD) and the tree metric. The new architecture is also empirically analyzed on a set of standard benchmarks, specifically long-range tasks, geared at understanding how FSW-GNN alleviates oversmoothing and oversquashing.

**Strengths:**

- The paper is generally well-motivated and easy to follow.
- Designing an MPNN that, unlike practically all existing architectures, can *simultaneously* differentiate all WL-distinguishable graphs is an intriguing research direction. This approach aligns well with recent lines of work aimed at understanding GNN embeddings and stability.
- I found the generalization of the DS metric to node-featured graphs particularly insightful, which I believe to be an important step towards establishing it further as a WL metric alongside the TMD etc.

**Weaknesses:**

- While the extension of the DS metric to node-featured graphs is an important step, to me this seems somewhat disconnected from the rest of the paper, considering that the proof of the lower Lipschitzness of FSW-GNN solely relies on the vector norms on the features being $\ell^1$, giving rise to piecewise linear functions.
- In its current stage, to me the contribution seems rather incremental on top of [2], which proposed the FSW multiset embeddings. The fact that using these injective multiset embeddings in a MPNN also yields WL-equivalent MPNNs, which is emphasized as one of the main results of the work, does not come surprising to me, and the brevity of the proof of Theorem 3.2 underlines this.
- In the analysis of the bi-Lipschitzness (Theorem 3.3), the obtained Lipschitz constants are not made explicit. The proof also seems rather generic to me, as it relies only on the set of all graphs up to a certain number of nodes being finite, and the underlying distance functions to be piecewise linear w.r.t. the features. With a similar reasoning, one could also argue that all ReLU MLPs are Lipschitz continuous -- however, the constant could be arbitrarily bad. As such, I can only see limited practical utility in such a statement.
- While the method is evaluated on a range of standard benchmarks, in my opinion the work is missing a careful smaller-scale evaluation, potentially on synthetic data, which validates the theory (i.e., if most instances of FSW-GNN are really WL-equivalent), and empirically investigates the lower Lipschitz constant of the method. I am also unsure about the ability of FSW-GNN to alleviate oversquashing. Here, I would define oversquashing as the difficulty that MPNNs have at exchanging messages over multiple hops by having to squash information from exponentially increasing neighborhoods into a fixed-size vector. I believe that this might only be tested insufficiently by the graph transfer task, as only the information out of a single node is important for the task. I am unsure how FSW-GNN would tackle this differently than a standard MPNN, considering that the underlying graph used for message passing and the vectors for representation of the features stay unchanged.

In my opinion, the scope of the work in its current form, especially considering that the Lipschitz constant of FSW-GNN is not further quantified, is not yet ready for a full conference paper. Nonetheless, I find this a very interesting topic and genuinely enjoyed reading the paper, and with the above weaknesses addressed, I believe that this would be a strong contribution.

[1] Yair Davidson and Nadav Dym. On the Hölder Stability of Multiset and Graph Neural Networks. arXiv preprint arXiv:2406.06984, 2024.

[2] Tal Amir and Nadav Dym. Fourier sliced-Wasserstein Embedding for Multisets and Measures, 2024. URL https://arxiv.org/abs/2405.16519.

**Questions:**

- Did you attempt to further quantify the Lipschitz constant of Theorem 3.3, and/or conducted experiments to study this?
- There are several typos and minor errors in the script, to the point that reading flow is sometimes a bit interrupted.  I encourage the authors to further proofread the script in order to improve the presentation. Some examples I found include the following:
   - In the definition of lower-Hölder continuity (lines 220-222), "*in expectation*" should either be removed or the correct notion as introduced in [1] should be described.
   - In lines 1044/1045, it might say $\boldsymbol{h}_{\tilde{G}}$ instead.

---

> ### Author Response · Authors · 2024-11-24
> **Response to Reviewer yDYZ (Part 1 of 2)**
>
> ### Response to Summary
>
> We thank the reviewer for the detailed and accurate summary of our work. We would like to clarify one detail: the Fourier-Sliced Wasserstein (FSW) embedding for multisets is not merely _almost injective_, but rather _fully_ injective and, moreover, bi-Lipschitz. These properties play a crucial role in ensuring the theoretical robustness and practical effectiveness of the proposed FSW-GNN architecture.
>
> ### Response to Weaknesses
>
> > While the extension of the DS metric to node-featured graphs is an important step, to me this seems somewhat disconnected from the rest of the paper, considering that the proof of the lower Lipschitzness of FSW-GNN solely relies on the vector norms on the features being l1, giving rise to piecewise linear functions.
>
> We only use the $l\_1$ norm in an intermediate step in the proof. Our results hold for any $\ell_p$-norm on the output Euclidean space, and any $p$-Wasserstein distance for $p\in[1,\infty]$. Our proof presents a novel technique to show that functions that are injective and piecewise linear are bi-Lipschitz, which we believe has applications outside the scope of this paper.
>
> > In its current stage, to me the contribution seems rather incremental on top of [2], which proposed the FSW multiset embeddings. The fact that using these injective multiset embeddings in a MPNN also yields WL-equivalent MPNNs, which is emphasized as one of the main results of the work, does not come surprising to me, and the brevity of the proof of Theorem 3.2 underlines this.
>
> We emphasize that the primary contribution of our work lies not in achieving WL-equivalence, which has already been achieved in previous works, but in establishing the stronger property of bi-Lipschitzness. Our model is the first to provide such a guarantee, marking a significant advancement in the theoretical understanding of graph neural networks.
>
> While it may seem a-priori intuitive that employing a bi-Lipschitz multiset aggregation should yield a bi-Lipschitz MPNN, proving this is a substantial theoretical challenge. Our proof relies on heavy machinery from the recent theory of $\sigma$-subanalytic functions [FSW], as well as the piecewise-linearity technique presented in Lemma 3.4.
>
> > In the analysis of the bi-Lipschitzness (Theorem 3.3), the obtained Lipschitz constants are not made explicit. The proof also seems rather generic to me, as it relies only on the set of all graphs up to a certain number of nodes being finite, and the underlying distance functions to be piecewise linear w.r.t. the features. With a similar reasoning, one could also argue that all ReLU MLPs are Lipschitz continuous -- however, the constant could be arbitrarily bad. As such, I can only see limited practical utility in such a statement.
>
> We agree that explicitly bounding the Lipschitz constants would add value to our analysis. However, it is important to note that until the recent work of [FSW], the existence of such constants had not been established even for multisets. Explicitly bounding these constants would require entirely different theoretical tools, and is beyond the scope of this work. Nevertheless, we believe that this does not detract from the theoretical significance of our contribution, as it establishes a foundational result that opens the door for future explorations into bounding these constants.
>
> To provide additional insight into the practical implications of our bi-Lipschitzness guarantee, we include an empirical evaluation of the bi-Lipschitz distortion incurred by our method and its competitors. Please refer to Figure 1 in the revised manuscript.
>
> > With a similar reasoning, one could also argue that all ReLU MLPs are Lipschitz continuous -- however, the constant could be arbitrarily bad. As such, I can only see limited practical utility in such a statement.
>
> We agree that the argument in our bi-Lipschitzness proof could be used to show that ReLU *MLPs* that are injective (as a function defined on vectors) are bi-Lipschitz. However, this argument _cannot_ be used to show that MPNNs with ReLU MLPs are bi-Lipschitz if they are injective (as functions defined on graphs).
>
> A direct implication of the results in [FWT] is that MPNNs with piecewise-linear aggregation functions are never injective and, consequently, do not satisfy the assumptions of Lemma 3.4, nor do they have the stronger property of bi-Lipschitzness. Furthermore, the results in [FWT] imply that MPNNs with smooth aggregation functions are also never bi-Lipschitz. This distinction is  strongly supported by our distortion evaluation experiment (Figure 1).

---

> > ### Author Response · Authors · 2024-11-24
> > **Response to Reviewer yDYZ (Part 2 of 2)**
> >
> > > While the method is evaluated on a range of standard benchmarks, in my opinion the work is missing a careful smaller-scale evaluation, potentially on synthetic data, which validates the theory (i.e., if most instances of FSW-GNN are really WL-equivalent), and empirically investigates the lower Lipschitz constant of the method.
> >
> > Thanks for this suggestion. We added Figure 1, which shows that our method yields considerably lower bi-Lipschitz distortion in practice in comparison with other methods.
> >
> > > I am also unsure about the ability of FSW-GNN to alleviate oversquashing. [...] I am unsure how FSW-GNN would tackle this differently than a standard MPNN, considering that the underlying graph used for message passing and the vectors for representation of the features stay unchanged.
> >
> > Our argument at this point is just that our method dramatically outperforms competing MPNNs on these graph transfer tasks (Figure 3), which are widely recognized benchmarks for the oversquashing problem.  Regardless of whether this conception of the community is completely correct, this experiment shows the potential in using FSW aggregations to alleviate oversquasing. It is possible that additional tools will be required for more challenging, yet undeveloped, adversarial oversquasing examples.
> >
> > > In my opinion, the scope of the work in its current form, especially considering that the Lipschitz constant of FSW-GNN is not further quantified, is not yet ready for a full conference paper. Nonetheless, I find this a very interesting topic and genuinely enjoyed reading the paper, and with the above weaknesses addressed, I believe that this would be a strong contribution.
> >
> > Given that the primary reason for your recommendation of rejection is the lack of explicit bounds for the bi-Lipschitz constants, we respectfully request that you reconsider in light of our explanation above regarding the current research landscape. As we noted, the theoretical understanding of such constants is still in its early stages, and even their existence in the easier domain of multisets was only recently established.
> >
> > ### Response to Questions
> >
> > We fixed these errors in the revised manuscript. Thanks! Additionally, if the paper is accepted, we will conduct a throrough proofreading before the camera ready submission.
> >
> > ### References
> >
> > [FWT] Amir, T., Gortler, S., Avni, I., Ravina, R., & Dym, N. (2024). Neural injective functions for multisets, measures and graphs via a finite witness theorem. Advances in Neural Information Processing Systems, 36.
> >
> > [FSW] Tal Amir and Nadav Dym. Fourier sliced-Wasserstein Embedding for Multisets and Measures, 2024. URL https://arxiv.org/abs/2405.16519.

---

> ### Comment · Reviewer_yDYZ · 2024-11-26
> **Response by Reviewer**
>
> I thank the authors for their rebuttal and their thoughtful responses to my review. I sincerely apologize for the inaccurate wording regarding the injectivity of FSW-GNN, and I have updated the review accordingly.
>
> **Re technical contributions:**
>
> > We only use the $\ell_1$ norm in an intermediate step in the proof
>
> I thank the authors for the explanation, and I understand that the Lipschitzness also holds for any other norm beyond $\ell_1$, relying on the basic fact that all norms on finite dimensional vector spaces are equivalent.
>
> > Our proof presents a novel technique to show that functions that are injective and piecewise linear are bi-Lipschitz, which we believe has applications outside the scope of this paper.
>
> I agree that this is a nice result, and acknowledge the effort it takes to rigorously prove this fact -- however intuitively, this seems immediately obvious to me, and without a quantitative lower bound that is somewhat related to the model, I maintain my opinion that the bi-Lipschitz bound is of limited relevance.
>
> > While it may seem a-priori intuitive that employing a bi-Lipschitz multiset aggregation should yield a bi-Lipschitz MPNN, proving this is a substantial theoretical challenge. Our proof relies on heavy machinery from the recent theory of
> $\sigma$-subanalytic functions [FSW], as well as the piecewise-linearity technique presented in Lemma 3.4.
>
> I am not sure I agree that this presents a "substantial theoretical challenge". From my understanding, the "heavy machinery" (i.e., $\sigma$-subanalytic functions, finite witness theorem) is only used *indirectly* through Theorem 4.1 of [1], and the proof of Theorem 3.2 (L1011-1017) is a rather straightforward *application* of said theorem. I kindly ask the authors to further elaborate on their specific technical contribution in this regard, should I have overlooked some particular difficulty.
>
> **Re oversquashing:**
>
> > Our argument at this point is just that our method dramatically outperforms competing MPNNs on these graph transfer tasks (Figure 3), which are widely recognized benchmarks for the oversquashing problem. Regardless of whether this conception of the community is completely correct, this experiment shows the potential in using FSW aggregations to alleviate oversquasing. It is possible that additional tools will be required for more challenging, yet undeveloped, adversarial oversquasing examples.
>
> I thank the authors for their response. Taking the definition of oversquashing from [2] (and as defined in the review), from my understanding the said graph transfer tasks are *necessary*, but not *sufficient* for a graph ML model to alleviate oversquashing (as there is only one message that has to be propagated through the graph). Nevertheless, I fully agree that conceptions in the community about this vary slightly, and I want to emphasize that this was a very minor comment.
>
> **Re experiments:**
>
> > We added Figure 1, which shows that our method yields considerably lower bi-Lipschitz distortion in practice in comparison with other methods.
>
> I thank the authors for adding this interesting experiment, and the results for FSW-GNN's empirical distortion rates indeed look promising.
>
> **Conclusion:**
>
> I thank the authors for their responses and particularly for adding the empirical distortion rate experiment in Figure 1, which adds to the validity of the proposed method. To acknowledge the effort put into this, I am raising my score. Yet, I maintain my perception that the theoretical contributions are limited and that the work is too incremental on top of [3] and too close to SortMPNN from [4] (as also brought up by reviewer GwhL) to justify a higher score. As said in the review, I encourage the authors to keep working on a more quantitative version of Theorem 3.3.
>
>
>
>
> [1] Tal Amir, Steven J. Gortler, Ilai Avni, Ravina Ravina, and Nadav Dym. Neural injective functions for multisets, measures and graphs via a finite witness theorem. In Thirty-seventh Conference on Neural Information Processing Systems, 2023.
>
> [2] Uri Alon and Eran Yahav. On the bottleneck of graph neural networks and its practical implications. arXiv preprint arXiv:2006.05205, 2020.
>
> [3] Tal Amir and Nadav Dym. Fourier sliced-Wasserstein Embedding for Multisets and Measures, 2024. URL https://arxiv.org/abs/2405.16519.
>
> [4] Yair Davidson and Nadav Dym. On the hölder stability of multiset and graph neural networks. arXiv preprint arXiv:2406.06984, 2024.

---

> > ### Author Response · Authors · 2024-11-26
> > **Response to Reviewer**
> >
> > Thanks for your response.
> >
> > **Regarding the technical difficulty of our proof.** We agree that our proof has a certain component of piecing together non-trivial but known results from different places: using the properties of the FSW embedding, the finite witness theorem, and the observation that the metrics we consider are bi-Lipschitz equivalent to piecewise-linear metrics. We believe that this combination of known results is also an important contribution.
> >
> > In addition, the proof includes an important novel technical part, which was not covered by previous work. Lemma 3.4, which shows that piecewise-linear semi-metrics with the same zero set, defined on compact sets, are equivalent. This result may seem trivial, but it is not. For example, when the domain is not compact, there exist counterexamples for which the Lemma statement does not hold.
> >
> > The difficulty of mathematical results is subjective. But just to give a comparison, the paper [1] was written by Radu Balan and his group, unquestionably a world expert on this type of questions. One of the main results in the paper is the bi-Lipschitzness of his sort embedding. The fact that the injectivity of the embedding implies bi-Lipschitzness is stated in Theorem 3.10, and its proof took four pages to prove. No bi-Lipschitz constants are given.  While we were obviously aware of this proof, and were influenced by it, we note that:
> >
> > 1. Our lemma 3.4 automatically proves Balan’s Theorem 3.10. Since the embedding and metric there are homogeneous, one can reduce to the unit cube, which is compact.
> > 2.  In this paper, we prove the bi-Lipschitzness of a much more complex embedding, and much more complex metric, defined on graphs rather than multisets.
> >
> > Finally, we agree that characterizing the bi-Lipschitz constants is the next step, but we feel that a proof of their existence is already important. The advantage of bi-Lipschitz embeddings vs. non-bi-Lipschitz embeddings for graphs is demonstrated in our experiments.
> >
> > Thank you for your attention. We would be happy to provide additional clarification if necessary.
> >
> > [1] Permutation Invaraint Representation with Applications to Graph Deep Learning ,Balan, Haghani and Sing

---

> ### Comment · Reviewer_yDYZ · 2024-11-28
> **Response by Reviewer**
>
> I thank the authors for their explanations regarding the technical difficulty of their proofs.
>
> I do agree that upon first glance, Theorem 3.10 from the referenced paper [1] appears to follow from Lemma 3.4, though I did not rigorously check the entire proof and I cannot rule out that there are no subtleties I might have missed.
>
> I further want to mention that in the theory of subanalytic sets, the inequality of Lemma 3.4 is ~~a special case of~~ *closely related to* the Łojasiewicz inequality (cf. Theorem 6.4 of [2]). In particular, the lemma states that the Łojasiewicz exponent is 1 in the special case of piecewise linear functions. Following the proof from [2], it seems to me that this would be fairly straightforward to deduce. While I did not find an exact statement of Lemma 3.4 in the textbooks that I looked at, the lemma still strikes me as too elementary and general for *not* being standard knowledge in fields like real analytic geometry or nonsmooth optimization.
> That said, I recognize the importance of including and rigorously proving this result for the paper's completeness and self-containedness. Yet, I feel that Lemma 3.4 alone does not qualify as a strong technical contribution; this is, of course, quite subjective.
>
> I will maintain my score.
>
> [2] Edward Bierstone and Pierre D. Milman. Semianalytic and subanalytic sets. Publications mathématiques de l’I.H.É.S., tome 67 (1988), p. 5-42.

---

> > ### Author Response · Authors · 2024-11-28
> > **Response to Reviewer**
> >
> > > I further want to mention that in the theory of subanalytic sets, the inequality of Lemma 3.4 is a special case of the Łojasiewicz inequality (cf. Theorem 6.4 of [2]).
> >
> > We’d like to clarify, for the benefit of the other reviewers, that our Lemma 3.4 is not “a special case of the Łojasiewicz inequality”, since Lemma 3.4 explicitly proves the inequality with power $r=1$, whereas the Łojasiewicz inequality only proves the _existence_ of such power — which is insufficient to imply bi-Lipschitzness. We believe having read the entirety of your answer that this is what you meant as well, but these particular words could be misleading.
> >
> > > [...] Yet, I feel that Lemma 3.4 alone does not qualify as a strong technical contribution; this is, of course, quite subjective.
> >
> > We stress that we are not claiming that our work would be accepted to a journal on real analytic geometry. This paper focuses on making graph neural networks provably bi-Lipschitz and demonstrating the benefits of this property experimentally. This was not done before, and is unlikely to be achieved by researchers focused solely on real analytic geometry.

---

> ### Comment · Reviewer_yDYZ · 2024-11-29
> **Response by Reviewer yDYZ**
>
> I thank the authors for their comment.
>
> > We believe having read the entirety of your answer that this is what you meant as well, but these particular words could be misleading.
>
> I apologize if my wording was unclear, and I have updated the text.
>
> > In addition, the proof includes an important novel technical part, which was not covered by previous work. Lemma 3.4, which shows that piecewise-linear semi-metrics with the same zero set, defined on compact sets, are equivalent. This result may seem trivial, but it is not. For example, when the domain is not compact, there exist counterexamples for which the Lemma statement does not hold.
>
> > We stress that we are not claiming that our work would be accepted to a journal on real analytic geometry. This paper focuses on making graph neural networks provably bi-Lipschitz and demonstrating the benefits of this property experimentally. This was not done before, and is unlikely to be achieved by researchers focused solely on real analytic geometry.
>
> I appreciate this explanation. I also think that significant theoretical novelty is not necessary in works with a strong empirical part, as long as the theoretical aspects support and enhance the empirical findings. However, I believe that in this particular case, the proof of Bi-Lipschitzness is a core part of the contribution. My intention was not to diminish its value but to express some skepticism about the claimed novelty of Lemma 3.4.

---

### Official Review · Reviewer_6Vb9 · 2024-11-02

**Soundness:** 3
**Presentation:** 3
**Contribution:** 2
**Rating:** 5
**Confidence:** 3

**Summary:**

The paper proposed FSW-GNN, which makes use of Fourier Sliced-Wasserstein (FSW) embedding and guarantees the bi-Lipschitz property of the output graph embeddings. Empirically, FSW-GNN is on par with MPNN on standard graph learning tasks, but it achieves superior performance on long-range tasks.

**Strengths:**

1. Bi-Lipchitzness is an interesting idea to understand the expressive power of MPNNs.

2. It is quite interesting to see that FSW-GNN is particularly strong for long-range tasks, meaning that enhancing Bi-Lipchitzness is particularly effective for long-range tasks, where standard GNNs fall short of.

**Weaknesses:**

Since bi-Lipschitz continuity is stated for graph embeddings, it is less clear from a theoretical perspective in what way it is connected with node embeddings and node-level tasks. In particular, the long-range tasks considered are node-level tasks. The intuition stated in line 216-236 is also based on the graph level: I can see that "deep GNNs are bad for graph level tasks, and as a result through improving bi-Lipschitz continuity, FSW-GNN makes it less bad for graph level tasks", but it is hard to see why this intuition explains why FSW-GNN makes it less bad for node level tasks.

**Questions:**

See above.

---

> ### Author Response · Authors · 2024-11-24
> **Response to Reviewer 6Vb9**
>
> ### Response to Weaknesses
>
> **On node-level vs. graph-level tasks:** We thank the reviewer for this question, and would like to clarify the connection between the graph-level and node-level output. The bi-Lipschitzness of our graph-level embedings necessarily implies bi-Lipschitzness for the individual output vertex features. Lack of bi-Lipschitzness in the latter would preclude bi-Lipachitzness for the former, both theoretically and practically, as the latter is derived from the former.
>
> The purpose of the long-range experiments was to empirically demonstrate that the output node features computed by our GNN exhibit better information retention compared to those produced by other architectures. This aligns well with our theory and strengthens the practical relevance of our method.

---

> > ### Comment · Reviewer_6Vb9 · 2024-11-25
> >
> > Thank you for your answer. I am still confused how bi-Lipschitzness can be used to explain about node-level tasks. Because bi-Lipschitzness is defined on the graph level. What do you mean by "bi-Lipschitzness for the individual output vertex features" (I could not find a definition for it in the paper) and how could one use it to infer about node-level performance?

---

> > > ### Author Response · Authors · 2024-11-27
> > > **Response to Reviewer**
> > >
> > > What we mean is bi-Lipschitz with respect to the tree distance between node-based computational trees defined in Definition B.3. This graph level tree distance is based on this node-level tree distance.  This distance (which is actually a semi-distance), will be preserved in the bi-Lipschitz sense by FSW-GNN. We have not included this result because the proof is essentially the same as the graph-level metric, and we did not want to over-complicate the paper. A similar approach was taken in the SortMPNN paper, where the node-level distance is only mentioned as an afterthought in the appendix (page 29 in that paper).
> > >
> > > That being said, after rethinking, we agree your comment is a good one. Since our long-range tasks address node level tasks, including a discussion of the node level metric, and a proof of bi-Lipschitzness, would be a good addition to the paper. We will add this to the camera ready version of the paper, should the paper be accepted.

---

### Official Review · Reviewer_WoLa · 2024-11-04

**Soundness:** 3
**Presentation:** 3
**Contribution:** 2
**Rating:** 6
**Confidence:** 4

**Summary:**

This paper introduces FSW-GNN, a novel bi-Lipschitz GNN that maintains stable distance-preserving embeddings through the FSW aggregation method. FSW-GNN outperforms traditional MPNNs on tasks requiring deep message-passing, effectively addressing issues like over smoothing and over squashing in long-range tasks.

**Strengths:**

1. The paper introduces the FSW-GNN, a novel graph neural network (GNN) model that achieves bi-Lipschitz continuity, which is a significant advancement over traditional MPNNs that lack such properties.
2. Empirical evaluations show that FSW-GNN performs competitively on standard graph tasks and excels in long-range tasks.

**Weaknesses:**

1. The FSW-GNN requires more complex operations, potentially increasing runtime compared to simpler GNN architectures like GCN and GIN.
2. FSW-GNN’s runtime is considerably higher for large datasets, which might limit its application in highly scalable scenarios.
3. While the paper provides proof for the bi-Lipschitz properties, it relies on empirical evidence and some conjecture to suggest that the model’s stability holds as depth increases.

**Questions:**

1. TMD requires proper features for every vertex, which can not be available in practical cases, where it fails. Can this method be extended in those cases where partial node features are missing?
2. What are the average node numbers and edge numbers of the dataset that are used in the experiments?
3. How does the FSW-GNN handle graphs with different structures, such as highly dense versus sparse graphs, in terms of both performance and computational efficiency?
4. How critical is the bi-Lipschitz property for real-world applications beyond synthetic tasks?
5. The paper focuses on long-range tasks that benefit from deep message passing. How does FSW-GNN perform on tasks involving smaller graphs or shallow networks, like the MUTAG dataset?
6. How sensitive is FSW-GNN to changes in hyperparameters compared to traditional MPNNs?

---

> ### Author Response · Authors · 2024-11-24
> **Response to Reviewer WoLa**
>
> ### Response to Weaknesses
>
> > While the paper provides proof for the bi-Lipschitz properties, it relies on empirical evidence and some conjecture to suggest that the model’s stability holds as depth increases.
>
> Thank you for this insightful question. First, we’d like to clarify that our bi-Lipschitz proof hold for abitrary depth. It is true that distortion increases with depth in our method. However, the key advantage of our approach is that the distortion grows much more slowly compared to other methods. This is due to the higher distortion incurred per iteration in methods that lack bi-Lipschitzness. In fact, methods that are not bi-Lipschitz are theoretically proven to have infinite distortion on some pairs of graphs, even after just one iteration.
>
> This distinction is clearly demonstrated in our empirical distortion evaluation (Figure 1). Even after 10 iterations, the distortion incurred by our method remains significantly lower than that of just one iteration of most competing methods. This highlights the stability of our approach and its robustness to increasing depth.
>
> ### Response to Questions
>
> > TMD requires proper features for every vertex, which can not be available in practical cases, where it fails. Can this method be extended in those cases where partial node features are missing?
>
> Yes, this method can be extended to cases with partial node features. One approach is to augment the vertex feature vectors with a 0-1 indicator that specifies whether the corresponding vertex has a feature. In cases where a feature is missing, a placeholder vector (e.g., a zero vector) can be used as the dummy feature.
>
> > What are the average node numbers and edge numbers of the dataset that are used in the experiments?
>
> We added these statistics in Appendix A, thanks for the suggestion.
>
> > How does the FSW-GNN handle graphs with different structures, such as highly dense versus sparse graphs, in terms of both performance and computational efficiency?
>
> **Computation wise:** FSW-GNN is designed to handle both dense and sparse graphs efficiently. In particular, our implementation leverages PyTorch’s support for sparse tensors, allowing us to process sparse graphs with significant computational savings. In terms of runtime, FSW-GNN is computationally more intensive than simple GNNs that utilize basic message aggregation functions such as sum pooling. Yet, our method takes only four times longer to run compared to these simpler methods. This additional computational cost is expected due to the more sophisticated message aggregation used in FSW-GNN.
>
> **Performance wise:** It can be that the two graphs in Table 4 where SortMPNN outperforms our method are exactly the two graphs with relatively high average degree. We added a comment discussing this to the main text.
>
> > How critical is the bi-Lipschitz property for real-world applications beyond synthetic tasks?
>
> This is an important question. Indeed the purpose of all our experiment is to address it. Our experiments suggest that our bi-Lipschitz architecture is very helpful for the transductive task, and for long range tasks.
>
> > The paper focuses on long-range tasks that benefit from deep message passing. How does FSW-GNN perform on tasks involving smaller graphs or shallow networks, like the MUTAG dataset?
>
> We added an experiment on MUTAG. FSW-GNN outperforms all other MPNNs we compared against. Note also that several graphs in the transductive learning tasks are also rather small.
>
> > How sensitive is FSW-GNN to changes in hyperparameters compared to traditional MPNNs?
>
> In our experiments, FSW-GNN demonstrated low sensitivity to changes in hyperparameters. For instance, doubling or halving the hidden feature dimension had very little effect on the results for the Actor and Cora datasets.

---

### Official Review · Reviewer_GwhL · 2024-11-04

**Soundness:** 3
**Presentation:** 4
**Contribution:** 3
**Rating:** 6
**Confidence:** 4

**Summary:**

The authors of this paper propose FSW-GNN, a new message-passing neural network that achieves bi-Lipschitz continuity while maintaining Weisfeiler-Lehman (WL) equivalence. Unlike typical MPNNs, which struggle with feature separation in Euclidean space, FSW-GNN provides bi-Lipschitz guarantees by leveraging the Fourier Sliced-Wasserstein (FSW) embedding for message aggregation.

**Strengths:**

- The authors did an exceptionally good job in presenting and structuring the paper, which I enjoyed reading. The way the authors introduce the related work throughout the paper is very nice and detailed, as it gives a clear message to the readers about the contributions and the relations with previous works.
- FSW-GNN is one of the first GNNs to offer bi-Lipschitz guarantees with respect to two significant WL-metrics: the DS metric and Tree Mover’s Distance.
- The experimental results are promising, as the proposed method outperforms in various datasets simple GNN baselines.

**Weaknesses:**

- My main concern with this paper is its close similarity to SortMPNN, as many ideas of the paper, such as the use of sorted message aggregation, are directly inspired by that work. While FSW-GNN introduces bi-Lipschitz guarantees, SortMPNN already established a similar approach to achieving Lipschitz properties in expectation. Furthermore, in the experimental results, SortMPNN actually outperforms FSW-GNN. The authors also did not provide a comparison with SortMPNN on some of the other datasets used. Therefore, besides that FSW-GNN is computationally cheaper than SortMPNN, it is not clear to me why someone should use FSW-GNN instead of SortMPNN.

- The authors argue that FSW-GNN excels in long-range tasks, however, the reported results in peptides-struct and peptides-func, which are considered long-range datasets, are not that good. How do the authors explain this?

**Questions:**

- Could the authors clarify the specific advantages of FSW-GNN over SortMPNN beyond computational efficiency? Given the performance gap on some benchmarks, what key factors would make FSW-GNN better than SortMPNN?

- Could the authors provide the experimental results for SortMPNN in the examined datasets?

- Could the authors elaborate on why FSW-GNN did not perform as expected on these long-range datasets, and how this aligns with the claim of improved long-range performance?

- Since the sorting operation is not differentiable, how do the authors handle this in the training process?

---

> ### Author Response · Authors · 2024-11-24
> **Response to Reviewer GwhL (Part 1 of 2)**
>
> ### Response to Strengths
>
> We appreciate the reviewer’s thoughtful feedback and are glad that you found our manuscript enjoyable and well-presented.
>
>
> ### Response to Weaknesses
>
> Firstly, as you suggested, we have added a comparison to SortMPNN on all datasets considered in the paper, and also on the MUTAG dataset and distortion experiment suggested by other reviewers. As we wrote in our comment to all reviewers, FSW-GNN outperforms SortMPNN in most tasks: 7 of 9 tasks in Table 4, peptide-struct, all of our synthetic oversquashing experiments, and the new MUTAG and distortion experiments. In contrast, SortMPNN outperforms FSW-GNN only on peptide-func and 2 out of 9 datasets in Table 4.
>
>
> **Regarding the similarity to Sort-MPNN**
>
> To address the similarities and distinctions between FSW-GNN and SortMPNN, we wish to highlight two critical differences:
> 1. **Handling variable neighborhood sizes:** To accommodate vertex neighborhoods of varying size, SortMPNN uses _padding_, which requires the specification of a parameter $d\_{\max}$ that upper-bounds the maximal neighborhood size in the input graphs. This necessitates a priori knowledge of the maximal vertex degree across all input graphs to be processed by the model. Such a requirement can pose significant limitations in real-world use cases where this information is unavailable or unpredictable. Moreover, SortMPNN’s runtime and memory complexity scale linearly with $d\_{\max}$, making the use of arbitrarily large $d\_{\max}$ impractical. Additionally, padding introduces artificial distortion in the embedding space, which can impair performance. This is evidenced in our new experiments, as seen in Figure 3.   In contrast, FSW-GNN treats vertex neighborhoods as uniform distributions - a natural and principled approach that eliminates the need for padding or prior knowledge of maximum neighborhood size. This makes FSW-GNN computationally more efficient and conceptually more generalizable.
>
> 2. **Applicability to weighted graphs:** SortMPNN is limited to handling simple graphs or, at best, multigraphs, as its padding-based approach is not suited for weighted graphs. FSW-GNN, on the other hand, is inherently applicable to weighted graphs due to its treatment of vertex neighborhoods as distributions. This further extends the practical applicability of our method, and is directly supported in our implementation.
>
> **Theoretical guarantees:** Our work is the first to provide a theoretical bi-Lipschitzness guarantee for a GNN that hold uniformly and deterministically for all inputs. In contrast, the guarantees provided in the SortMPNN paper require a probabilistic assumption on the input, and hold only in expectation.  Furthermore, our novel proof technique, which combines the Finite Witness Theorem and piecewise-linearity argument of Lemma 3.4, can be extended to more general settings. We thus believe this technique is of independent interest and has the potential to support future research beyond the scope of this work.
>
> > The authors argue that FSW-GNN excels in long-range tasks, however, the reported results in peptides-struct and peptides-func, which are considered long-range datasets, are not that good. How do the authors explain this?
>
> We thank the reviewer for this question. Indeed, the goal of designing LRGB (Long-Range Graph Benchmark) was to design a dataset of long-range tasks on which graph transformers could outperform MPNN. Note that, in the language of the original paper [LRGB], the dataset “is not directed at proposing ‘provable LRI (Long Range Interaction)' benchmarks, which would often lead to toy datasets” but rather on proposing real-world tasks which are believed to be long range. The flip side of this is that we are not completely sure that these tasks are indeed long range. Indeed, regarding the two peptide datasets considered in this paper, the reasoning given for the long-rangeness is rather circumstantial: “Thus, the proposed Peptides-func and Peptides-struct are better suited to benchmarking of graph Transformers or other expressive GNNs, as they contain larger graphs, more data points, and challenging tasks.”
>
> Additionally, a later paper [GAP] showed that, with appropriate hyperparameter tuning, MPNNs with depth of 6-10 can achieve very competitive results, in comparison with graph transformers. These are the results we compare with in our paper. The success of standard MPNN of reasonable depth casts some doubt on the long-rangeness of Peptides-Struct and Peptides-Func.
>
> In contrast, the long range datasets we consider in this paper, Figures 2-3, are popular synthetic datasets, but are provably long range: an MPNN with less depth than the problem radius will never be successful.

---

> > ### Author Response · Authors · 2024-11-24
> > **Response to Reviewer GwhL (Part 2 of 2)**
> >
> > ### Response to Questions
> >
> > Question 1-3 are addressed in response to Weaknesses.
> >
> > > Since the sorting operation is not differentiable, how do the authors handle this in the training process?
> >
> > The sort function is a continuous, piecewise-linear function (same properties as ReLU networks), and differentiation is provided by pytorch’s sort operation. This is what we use. It is true that there are other variants of the sorting operation that are not continuous and piecewise differentiable, like lexicographical sort (which is piecewise continuous) or argsort (which is piecewise constant).
> >
> > ### References
> >
> > [LRGB] Dwivedi, Vijay Prakash, et al. "Long range graph benchmark." Advances in Neural Information Processing Systems 35 (2022): 22326-22340.
> >
> > [GAP] Tönshoff, Jan, et al. "Where did the gap go? reassessing the long-range graph benchmark." arXiv preprint arXiv:2309.00367 (2023).
> >
> > [FWT] Amir, T., Gortler, S., Avni, I., Ravina, R., & Dym, N. (2024). Neural injective functions for multisets, measures and graphs via a finite witness theorem. Advances in Neural Information Processing Systems, 36.

---

> > > ### Comment · Reviewer_GwhL · 2024-11-26
> > >
> > > I would like to thank the authors for answering my questions. I believe the paper presents interesting results, however I agree with reviewer yDYZ, as most of the contributions are incremental on top of [3,4]. Therefore, I will retain my initial score.
> > >
> > >
> > >
> > >
> > > Ref:
> > >
> > > [3] Tal Amir and Nadav Dym. Fourier sliced-Wasserstein Embedding for Multisets and Measures, 2024. URL https://arxiv.org/abs/2405.16519.
> > >
> > > [4] Yair Davidson and Nadav Dym. On the hölder stability of multiset and graph neural networks. arXiv preprint arXiv:2406.06984, 2024.

---

### Author Response · Authors · 2024-11-24
**Response to All Reviewers**

We thank the reviewers for their valuable feedback. We have uploaded a new version of the paper, with changes highlighted in blue. This new version includes several improvements based on your suggestions:

* We added SortMPNN to our experiments, as suggested by reviewer GwhL.

* We added an experiment on the MUTAG dataset, as suggested by Reviewer WoLa. As seen in Table 2, our FSW-GNN outperforms all other MPNNs (including Sort-MPNN).

* We added an experiment to evaluate the bi-Lipschitz distortion of MPNNs, as suggested by  Reviewer yDYZ. The results, appearing on Figure 1, clearly show that FSW-GNN has significantly better bi-Lipschitz distortion than other MPNNs, including SortMPNN.

* FSW-GNN outperforms SortMPNN in most tasks:    peptide-struct (Table 3), 7 of 9 tasks in Table 1, and all or our synthetic oversquashing experiments. This is also the case in the MUTAG and distortion experimentes mentioned above. In contrast, SortMPNN outperforms FSW-GNN only on peptide-func and 2 out of 9 datasets in Table 4.

* We added statistics of the datasets used (average degree etc.), as suggested by reviewer WoLa; please see Appendix A.

---

### Meta-Review · Area_Chair_tBDR · 2024-12-21

**Metareview:**

This paper proposes a GNN with the same expressive power as MPNN which is bi-Lipschitz with respect to the input.

The reviewers appreciated the paper's presentation and the numerical evaluation, especially after the revision where more experiments were added. However, some reviewers emphasized that the results were incremental over existing work [1,2].

[1] Tal Amir and Nadav Dym. Fourier sliced-Wasserstein Embedding for Multisets and Measures, 2024. URL https://arxiv.org/abs/2405.16519.

[2] Yair Davidson and Nadav Dym. On the hölder stability of multiset and graph neural networks. arXiv preprint arXiv:2406.06984, 2024.

**Additional Comments On Reviewer Discussion:**

Reviewer GwhL gave a score of 6 and was satisfied by the author's response but ended the discussion period saying that most results are incremental with respect to existing work.

Reviewer WoLa gave an initial score of 6 and did not engage in further discussions. The authors' response was satisfactory, especially regarding the inclusion of new numerical experiments. The reviewer's concern about the computational complexity of the new method seems to be a valid one.

Reviewer 6Vb9 gave a score of 5 and the weaknesses they mentioned were addressed by the authors. They did not change their scores.

Reviewer yDYZ reviewed the theorems carefully and gave a score of 5. The main weakness they found is the overlap and incremental contribution over existing work. They had a back-and-forth discussion with the authors regarding the difficulty of the proofs.

---

### Decision · Program_Chairs · 2025-01-22

Reject